# Comparative Study of the Steroidogenic Effects of Human Chorionic Gonadotropin and Thieno[2,3-D]pyrimidine-Based Allosteric Agonist of Luteinizing Hormone Receptor in Young Adult, Aging and Diabetic Male Rats

**DOI:** 10.3390/ijms21207493

**Published:** 2020-10-11

**Authors:** Andrey A. Bakhtyukov, Kira V. Derkach, Maxim A. Gureev, Dmitry V. Dar’in, Viktor N. Sorokoumov, Irina V. Romanova, Irina Yu. Morina, Anna M. Stepochkina, Alexander O. Shpakov

**Affiliations:** 1Department of Biochemistry, I.M. Sechenov Institute of Evolutionary Physiology and Biochemistry of Russian Academy of Sciences, 194223 Saint Petersburg, Russia; bahtyukov@gmail.com (A.A.B.); derkatch_k@list.ru (K.V.D.); irinaromanova@mail.ru (I.V.R.); irinamorina@mail.ru (I.Y.M.); annastepochkina23.11@mail.ru (A.M.S.); 2Department of Bioinformatics and Medicinal Chemistry, I.M. Sechenov First Moscow State Medical University, 119991 Moscow, Russia; max_technik@mail.ru; 3Department of Organic Chemistry, Institute of Chemistry, Saint Petersburg State University, 198504 Saint Petersburg, Russia; d.dariin@spbu.ru (D.V.D.); sorokoumov@gmail.com (V.N.S.)

**Keywords:** low-molecular-weight agonist, luteinizing hormone receptor, human chorionic gonadotropin, steroidogenesis, spermatogenesis, diabetes mellitus, aging rats, testes

## Abstract

Low-molecular-weight agonists of luteinizing hormone (LH)/human chorionic gonadotropin (hCG) receptor (LHCGR), which interact with LHCGR transmembrane allosteric site and, in comparison with gonadotropins, more selectively activate intracellular effectors, are currently being developed. Meanwhile, their effects on testicular steroidogenesis have not been studied. The purpose of this work is to perform a comparative study of the effects of 5-amino-*N*-*tert*-butyl-4-(3-(1-methylpyrazole-4-carboxamido)phenyl)-2-(methylthio)thieno[2,3-*d*] pyrimidine-6-carboxamide (TP4/2), a LHCGR allosteric agonist developed by us, and hCG on adenylyl cyclase activity in rat testicular membranes, testosterone levels, testicular steroidogenesis and spermatogenesis in young (four-month-old), aging (18-month-old) and diabetic male Wistar rats. Type 1 diabetes was caused by a single streptozotocin (50 mg/kg) injection. TP4/2 (20 mg/kg/day) and hCG (20 IU/rat/day) were administered for 5 days. TP4/2 was less effective in adenylyl cyclase stimulation and ability to activate steroidogenesis when administered once into rats. On the 3rd–5th day, TP4/2 and hCG steroidogenic effects in young adult, aging and diabetic rats were comparable. Unlike hCG, TP4/2 did not inhibit LHCGR gene expression and did not hyperstimulate the testicular steroidogenesis system, moderately increasing steroidogenic proteins gene expression and testosterone production. In aging and diabetic testes, TP4/2 improved spermatogenesis. Thus, during five-day administration, TP4/2 steadily stimulates testicular steroidogenesis, and can be used to prevent androgen deficiency in aging and diabetes.

## 1. Introduction

One of the pharmacological approaches for correcting androgen deficiency and impaired spermatogenesis in male patients with hypogonadotrophic hypogonadism and infertility is the use of gonadotropins, such as luteinizing hormone (LH) and its structural and functional homologue, human chorionic gonadotropin (hCG) [1,2,3,4]. In Leydig cells of the testes, LH and hCG with high affinity bind to a large extracellular domain (ectodomain) of LH/hCG receptor (LHCGR) belonging to the G protein-coupled receptors (GPCRs) superfamily, thereby causing the activation of heterotrimeric G-proteins (G_s_, G_q/11_) and β-arrestins [5,6,7,8]. The main mechanism of gonadotropin stimulation of steroidogenesis in Leydig cells is the G_s_-mediated activation of the enzyme adenylyl cyclase (AC), which causes an increase in intracellular cyclic adenosine monophosphate (cAMP) level. An increase in the cAMP level leads to the activation of protein kinase A, which stimulates the activity and expression of a wide range of steroidogenic proteins, and also activates the type 8 cAMP-specific phosphodiesterase, hydrolyzing cAMP and thereby regulating the testicular steroidogenesis [6,7,9,10]. In Leydig cells, the most important targets of protein kinase A are the steroidogenic acute regulatory protein (StAR), a mitochondrial cholesterol-transporting protein, and cytochrome P450_scc_ (CYP11A1) catalyzing pregnenolone synthesis (Appendix A) [11,12], as well as the sterol-regulatory element-binding protein (SREBP) cleavage-activating protein (SCAP)/SREBP pathway [10].

At the same time, the clinical use of gonadotropin preparations leads to a number of undesirable effects due to their heterogeneity (especially in the case of urinary hCG), low selectivity for intracellular effectors systems and their ability to induce resistance of target cells to endogenous LH [6,7,13,14,15]. The gonadotropin-induced activation of cAMP-independent signaling pathways is involved in the regulation of growth, differentiation, and cytoskeletal reorganization, as well as LHCHR desensitization in the target cells [6,7].

The low-molecular-weight (LMW) agonists of LHCGR are possibly a good alternative to gonadotropins. Unlike LH and hCG, they bind to the allosteric site, which is localized inside the LHCGR transmembrane domain formed by seven hydrophobic helices [16,17]. Of greatest interest among LHCGR allosteric agonists are the thieno[2,3-*d*]pyrimidine (TP) derivatives, which, as shown earlier by us and other authors, activate steroidogenesis in in vitro and in vivo conditions [16,18,19,20,21,22,23]. The TP derivatives are more selective as the regulators of intracellular signaling in target cells in comparison with gonadotropins. The compound Org 43553 (Appendix A), discovered by Dutch scientists in 2002, and the compounds TP1 and TP3, developed by us, selectively activate the G_s_ proteins and AC, but have a little effect on the G_q/11_ proteins and intracellular calcium signaling [16,24].

Dysfunctions in the hypothalamic-pituitary-gonadal axis, which are common in male patients with type 1 diabetes mellitus (T1DM) and in older men, lead to an imbalance in steroid hormones, androgen deficiency and impaired spermatogenesis [25,26,27,28,29]. The main cause for this is the impaired steroidogenic response of the testes to gonadotropins [30]. It is shown that the gonadotropin stimulating effect on androgen production in T1DM patients is reduced, especially in severe and moderate forms of this disease [28,29,31]. In turn, older men with increased LH levels often exhibit testicular resistance to gonadotropins and androgen deficiency [32,33,34]. In T1DM and aging, a decrease in testicular sensitivity to LH and hCG may be due to both a decrease in the number of functionally active LHCGRs and impaired LH signaling in Leydig cells, as a result of increased oxidative stress, apoptosis and inflammation in the testes [30,35].

Since the molecular mechanisms of interaction of allosteric agonists with LHCGR are significantly different from those of gonadotropins, we hypothesized that the effects of gonadotropins and TP derivatives on testicular steroidogenesis and spermatogenesis may also differ. This assumption is based on our early data that TP3 causes LHCGR desensitization to a much lesser extent in comparison with hCG [24,36]. In addition, being hydrophobic substances, TP derivatives are able to penetrate into the cell, bind to the “immature” forms of LHCGR and facilitate their translocation into the plasma membrane, thereby functioning as LMW chaperones [37]. This is very important in the case of T1DM and aging, where inflammation, oxidative stress and endoplasmic reticulum stress are triggered in the testes, and this leads to the damage of testicular proteins and impairs their post-translational modification and transport [38,39,40,41].

Based on the above, the aim of our work was to carry out a comparative study of the effects of 5-amino-*N*-*tert*-butyl-4-(3-(1-methylpyrazole-4-carboxamido)phenyl)-2-(methylthio)thieno[2,3-*d*] pyrimidine-6-carboxylic acid amide (TP4/2) (Appendix A) and hCG on AC activity in testicular membranes in the in vitro conditions, as well as on testosterone (T) levels, testicular steroidogenesis and morphology of the seminiferous tubules in young adult (four-month-old), aging (18-month-old) and diabetic male Wistar rats with a five-day administration of these drugs. We have shown for the first time the significant differences between the steroidogenic effects of TP4/2, an allosteric agonist of LHCGR, and hCG, an orthosteric agonist of this receptor, with their single and long-term (five-day) administration. The steroidogenic effect of TP4/2 after its single administration into young, aging and diabetic male rats was significantly less pronounced than that of hCG, and this was in a good agreement with the lower efficiency of TP4/2 in the in vitro experiments as a stimulator of AC activity in rat testicular membranes. At the same time, on the 3rd–5th days, the steroidogenic effect of hCG after the peak reached on the first and second days was reduced, while the corresponding effect of TP4/2 was preserved and became comparable to that of hCG. We showed that, in contrast to hCG, long-term administration of TP4/2 moderately stimulated the expression of steroidogenic genes encoding the StAR protein and cytochrome CYP17A1, but had little effect on LHCGR gene expression, which was significantly reduced upon hCG treatment. This is the first evidence that LMW allosteric agonists of LHCGR, inducing the steroidogenic effect comparable to that of gonadotropin, have a lesser effect on the pattern of steroidogenic proteins and do not suppress LHCGR gene expression, which is important for maintaining the testes’ sensitivity to endogenous gonadotropins. Most importantly, these effects were detected in the conditions of impaired testicular function in aging and T1DM. For the first time, it was shown that LHCGR allosteric agonist is capable of restoring the maturation of germ cells, impaired in aging and T1DM. This indicates the prospects for the use of LHCGR allosteric agonists for improving spermatogenesis in pathological conditions.

## 2. Results

### 2.1. The Metabolic and Hormonal Parameters in Diabetic and Aging Rats

We studied young adult (four-month-old) and aging (18-month-old) male rats and the animals with 30-day streptozotocin-induced T1DM that were treated with saline (s.c.), dimethyl sulfoxide (DMSO) (i.p.), TP4/2 (20 mg/kg/day, i.p.) or hCG (20 IU/rat/day, s.c.) for five days (for details see the Materials and Methods and Appendix A). In aging rats, increased body weight and reduced GSI were shown, but there were no significant changes in the hormones and glucose levels (Table 1). In diabetic rats, HbA1c content and glucose levels were increased, and the testes weight and the blood levels of insulin and leptin were decreased compared to the control (Table 1). Five-day treatment of young adult, aging and diabetic rats with TP4/2 and hCG did not significantly affect the HbA1c content and the blood levels of glucose, insulin and leptin. Simultaneously, hCG and, to a lesser extent, TP4/2 led to the inhibition of LH production, reducing the blood LH level (Table 1). The treatment of animals with hCG, but not with TP4/2, led to a significant increase in the GSI in diabetic rats and the testes weight in young adult and aging rats (Table 1). It should be noted that the administration of DMSO as a solvent did not affect the estimated parameters in the studied series of animals (Table 1).

### 2.2. The Effects of TP4/2 and hCG on AC Activity in Testicular Membranes of the Young Adult, Aging and Diabetic Rats

In the testicular membranes of diabetic rats, the basal and forskolin- and 5′-guanylylimidodiphosphate (GppNHp)-stimulated AC activities were decreased, while in aging rats, only GppNHp-stimulated AC activity was reduced significantly (Appendix A). The AC stimulating effects of TP4/2 and hCG were reduced in both aging and diabetic rats, to the greatest extent in the case of hCG in the membranes isolated from diabetic animals (Figure 1). This was illustrated by a decrease in the maximal AC stimulating effects of TP4/2 and hCG, as well as by an increase in the EC_50_ values for the effects of TP4/2 (601 ± 112 nM in young rats to 1006 ± 276 and 1517 ± 424 nM in the aging and diabetic groups, respectively, *p* < 0.05 as compared to young animals) and hCG (0.248 ± 0.045 nM in young rats to 0.398 ± 0.062 and 0.331 ± 0.053 nM in the aging and diabetic groups, respectively, *p* < 0.05 as compared to young animals) (Figure 1).

### 2.3. The Effect of a Single Administration of TP4/2 and hCG on the Blood Levels of Testosterone in the Young Adult, Aging and Diabetic Rats

A single treatment of young adult rats with hCG led to an increase in blood T level, which was more pronounced compared to rats treated with TP4/2. In the Young+hCG group, the maximal stimulating effect of gonadotropin was achieved 3 h after treatment (*p* < 0.001), and then began to decrease, while in the Young+TP group this effect was developed more slowly and did not reduce after 5 h (*p* < 0.05) (Figure 2).

In aging and diabetic rats, basal T level was significantly reduced compared to young adult animals, to the greatest extent, in T1DM (Figure 2). In aging and diabetes, the stimulating effects of TP4/2 and hCG were decreased in comparison with the Young+TP and Young+hCG groups, as demonstrated by the reduced AUC_10.00–15.00_ value for the “T concentration (nM)–time (hours)” curves (Figure 2D). The ratio of stimulating effects of TP4/2 and hCG in the studied groups was changed at different time points. In the young adult and diabetic rats, an hour after drug administration, TP4/2-induced an increase in the T level was, respectively, four and five times lower than in the case of hCG treatment (*p* < 0.001), while in aging rats, the stimulating effects of TP4/2 and hCG do not differ as much (*p* < 0.05) (Figure 2). After 3 h, in all the studied groups, the stimulating effects of TP4/2 were 33–41% of those of hCG. After 5 h, in the Young+TP, Aging+TP and Diab+TP groups, the effects of TP4/2 were 54, 54 and 64% of those of hCG (Figure 2).

### 2.4. The Effect of TP4/2 on the AC Activity in Thyroidal Membranes and the Thyroid Hormones Levels

Since the structure of the allosteric site of LHCGR is similar to that of thyroid stimulating hormone receptor (TSHR), we studied the effect of TP4/2 on the AC activity in the rat thyroidal membranes in in vitro conditions and on the baseline and thyrotropin-releasing hormone (TRH)-stimulated blood levels of thyroid hormones in young adult rats in in vivo conditions. In the in vivo experiments, TRH was administered intranasally (300 μg/kg), as previously described [42].

No significant effect of TP4/2 (10^−5^ M) on the basal and forskolin-, GppNHp- and TSH-stimulated AC activity in the rat thyroidal membranes was found (Appendix A). With a single administration to rats, TP4/2 at a dose of 20 mg/kg had no effect on the baseline and TRH-stimulated blood levels of free and total thyroxine and total triiodothyronine within 3.5 h after TP4/2 administration (Appendix A).

### 2.5. The Effect of Five-Day Treatment of the Young Adult, Aging and Diabetic Rats with TP4/2 and hCG on the Blood Testosterone Level and the Intratesticular Content of Testosterone and Its Precursors

During five days of administration, in young adult rats, the stimulating effect of TP4/2 on blood T levels remained unchanged (Figure 3A). On the first and second days, the corresponding effect of hCG was maximal and, on average, 2.5 times higher than that of TP4/2. On the third, fourth and fifth days, this effect was decreased and did not differ from that of TP4/2 (*p* > 0.05) (Figure 3A). The calculated AUC_0–5_ values for TP4/2 and hCG stimulating effects were significantly higher than in untreated groups, and the AUC_0–5_ for hCG effect was 58% higher than that for TP4/2 effect (Figure 3D). The ratio between stimulating effects of TP4/2 and hCG increased from 0.36 on the first day to 0.80 on the fifth day, which indicates a decrease in the differences in the effectiveness of these drugs in the case of five-day administration (Figure 3E). Five-day treatment with TP4/2 and hCG increased the intratesticular content of 17-hydroxyprogesterone, androstenedione and T in young adult animals (Table 2). The intratesticular 17-hydroxyprogesterone content in the group Young+hCG was significantly higher than that in the group Young+TP, while the content of T and other hormones in these groups did not differ significantly (Table 2).

The stimulating effects of TP4/2 and hCG on the blood T levels during their five-day administration to the aging and diabetic rats were reduced in comparison with the young adult series, as illustrated by the AUC_0–5_ values (Figure 3D). In aging and T1DM, TP4/2 effects were comparable and reached a plateau on the second day of treatment (Figure 3B,C). On the 3rd–5th days, the effects of TP4/2 and hCG on the blood T levels in the aging and diabetic rats were similar (Figure 3B,C). In the aging and diabetic rats, as in the young adult series, the ratio of the stimulating effects of TP4/2 and hCG on the blood TP levels was increased during their five-day administration (Figure 3E).

In the aging and diabetic rats, the stimulating effects of hCG and TP4/2 on the content of 17-hydroxyprogesterone and T in the testes were lower than in the young adult groups, and in the diabetic rats the stimulating effects of hCG and TP4/2 on the content androstenedione were reduced (Table 2). The intratesticular T content in the Diab+hCG and Diab+TP groups was similar, and the difference in intratesticular T content between the Aging+hCG and Aging+TP groups was weak (Table 2) and correlated well with the blood T levels (Figure 3B). Quite unexpected was the accumulation of progesterone in the aging series, especially in the group Aging+hCG (Table 2).

### 2.6. The Effect of Five-Day Treatment of Male Rats with TP4/2 and hCG on Intratesticular Expression of Genes Encoding LHCGR, StAR and Steroidogenic Enzymes

In the testes of aging and diabetic animals, the expression of gene *Lhr* encoding LHCGR tended to decrease, but the difference from the young adult rats was not significant (Figure 4). Five-day treatment of young adult rats with TP4/2 led to an increase in the *Lhr* expression and did not affect this expression in the groups Aging+TP and Diab+TP. At the same time, hCG induced a decrease in the *Lhr* expression in all the studied groups, to the greatest extent in T1DM (Figure 4).

Furthermore, the gene expression of StAR and four main steroidogenic enzymes catalyzing the synthesis of pregnenolone (cytochrome P450_scc_, CYP11A1, gene *Cyp11a*), progesterone (3β-hydroxysteroid dehydrogenase, gene *Hsd3b*), 17-hydroxyprogesterone, androstenedione (cytochrome P450 17A1/steroid 17α-monooxygenase, CYP17A1, gene *Cyp17a*), and testosterone (3β-hydroxysteroid dehydrogenase, gene *Hsd17b*) (for more details, see Appendix A) was studied.

In aging and T1DM animals, the basal expression of genes encoding steroidogenic proteins did not change significantly (Figure 4). In young adult animals, five-day treatment with TP4/2 led to an increase in the *Star* and *Cyp17a* expression, while hCG significantly increased the expression of the *Star*, *Cyp11a* and *Cyp17a* (Figure 4). In aging and diabetic rats, the stimulating effects of TP4/2 and hCG on the *Star* and *Cyp17a* expression were decreased, except for TP4/2′s stimulating effect on the *Star* expression in diabetic rats (Figure 4). Meanwhile, hCG stimulating effect on the *Cyp11a* expression was reduced only in T1DM (Figure 4). The *Hsd3b* gene expression was increased significantly in the groups Aging+hCG, Diab+TP and Diab+hCG, and the *Hsd17b* gene expression was slightly increased in the group Diab+TP. In the young adult animals, TP4/2 and hCG did not affect the gene expression of both dehydrogenases (Figure 4).

### 2.7. The Effect of TP4/2 and hCG Treatment on the Thickness of the Seminiferous Epithelium and the Number of Germ Cells within the Seminiferous Tubules

To study the influence of TP4/2 and hCG on the morphology of the seminiferous tubules and the number of germ cells, the fixed sections of the testes were stained with hematoxylin/eosin (Figure 5, Figure 6 and Figure 7). In the young adult rats, TP4/2 increased the number of pachytene spermatocytes, while hCG increased the number of both spermatogonia and spermatocytes (Table 3, Figure 5). In the aging and diabetic rats, a decrease in the thickness of the seminiferous epithelium and a significant decrease in the number of spermatogonia and spermatocytes were shown (Table 3, Figure 6 and Figure 7). Both TP4/2 and hCG caused the restoration of the number of germ cells in the seminiferous tubules of aging and diabetic rats to control values (Table 3, Figure 6 and Figure 7). In the Diab+TP group, the number of spermatogonia and spermatocytes was higher than in the young adult rats (Table 3). In aging and diabetic rats, both drugs restored the thickness of the seminiferous epithelium (Table 3).

### 2.8. Rat LHCGR and TSHR Transmembrane Allosteric Site Modelling and Molecular Docking of Thienopyrimidine Derivatives

A comparative analysis of the TP4/2 and Org 43553 (Appendix A) interaction with LHCGR allosteric site demonstrated the comparable GlideScore and Emodel values (Table 4). The most significant parameter is the hydrophobic contact intensity, described as the Lipo value, and less significant are coulombic interactions («Coulomb») and hydrogen bonding (Table 4). The GlideScore is a scoring-function [43] including binding free energy components for the ligands TP4/2 and Org 43553 in the allosteric site of LHCGR. The Emodel value combines the energy grid score, the binding affinity predicted by GlideScore, and the internal strain energy for the model potential used to direct the conformational-search algorithm which leads to a stable docking solution, considered as a potential ligand-receptor complex. Resulting values showed the reference parameters for LHCGR-specific compound Org 43553 and ligand TP4/2 docked in the LHCGR allosteric site (Table 4). Calculated values allow us to note the importance of hydrophobic contacts, which play a key role in the specific protein–ligand or protein–protein recognition [44]. The LHCGR allosteric site occupied by the TP derivatives is shown in Figure 8. Lipo values for both the studied compounds have significant negative values, and this indicates the preference of lipophilic ligand–receptor interactions for them (Table 4).

In the case of TSHR, the Org 43553 and TP4/2 docked into the allosteric site of the receptor showed affinity loss, expressed in raised GlideScore/Emodel values and also had a decreased increment of negative Lipo value, which indicates the loss of site-specific hydrophobic contacts towards transmembrane ligand-binding site (Table 4). Based on this fact, we can conclude that TP4/2 is potentially selective to the LHCGR allosteric site and with a low probability is able to interact with the allosteric site of TSHR, which is well supported by our experimental data on TP4/2′s effect on testicular steroidogenesis and thyroid hormones levels in rats (Figure 2 and Figure 3, Table 2, and Appendix A).

## 3. Discussion

The orthosteric site of LHCGR is localized in its large extracellular domain, and LH and hCG bind to it with high affinity, activating various types of heterotrimeric G-proteins and β-arrestins [5,6,7]. LHCGR also contains the allosteric sites that are located in both the intracellular and extracellular regions and within the transmembrane domain formed by seven hydrophobic helices [16,17,45]. It should be noted that, in a large number of GPCRs, a high-affinity orthosteric site is located within the transmembrane domain. It can be assumed that in evolutionary precursors of the LHCGR and other receptors of pituitary glycoprotein hormones, the transmembrane orthosteric site, due to structural changes in the transmembrane domain, including the replacement of the proline residue in the fifth transmembrane region, which is highly conserved in Class A GPCRs [46,47,48], has lost the ability to bind hormonal regulators and, as a result, transformed into an allosteric site. It can be assumed that in the course of evolution, in LHCGR and structurally related TSHR and follicle-stimulating hormone receptor, the transmembrane orthosteric site has lost the ability to bind hormones and converted to an allosteric site. Despite the fact that during activation by gonadotropins the LHCGR allosteric site remains free, it ensures the transfer of conformational rearrangements from the extracellular ligand-bound domain to the intracellular loops, and provides an active conformation or a set of active conformations of gonadotropin-activated LHCGR, which is required for effective interaction with heterotrimeric G-proteins and β-arrestins [6,7].

The low specificity of LH and hCG preparations to the LH-regulated signaling cascades and the possibility of their hyperactivation under conditions of using high-dose gonadotropins are the main reasons for the undesirable effects of LH and hCG when used in the clinic [6,7,13,14]. In this regard, over the past two decades, the search and development of allosteric regulators of the LHCGR, which would be devoid of these disadvantages, have been carried out. As a result, heterocyclic compounds with the activity of allosteric full and inverse agonists and allosteric neutral antagonists of LHCGR have been developed and studied, and they specifically bind to the transmembrane allosteric site of the receptor [49,50,51]. Based on the structure–activity relationship of LMW agonists of LHCGR, a suitable three-dimensional model for the LHCGR allosteric site was built [52]. The most active among LHCGR allosteric agonists is Org 43553, which belongs to the class of TPs [16,18,19,20]. Based on this, we earlier developed a series of the TP derivatives, among which the compounds TP1 and TP3 were more active in the in vitro and in vivo conditions [21,22,23,36]. In the present work, with the use of molecular modelling, we designed the compound TP4/2, interacting with the transmembrane allosteric site of LHCGR, with suitable parameters (Table 4, Figure 8). Our experimental results confirm the ability of TP4/2 to stimulate LHCGR-dependent testicular steroidogenesis (Figure 2 and Figure 3, Table 2).

Despite the functional differences, the structural organization of the transmembrane allosteric sites LHCGR and TSHR have features of similarity [53]. As a result, it is important to exclude the effect of LHCGR allosteric agonists on TSHR activity. Earlier, it was shown that Org 41841, which belongs to the TP derivatives, and a number of other LMW agonists of LHCGR are able to affect, albeit weakly, TSHR activity, which made it possible to create selective allosteric regulators of TSHR [52,53,54]. In contrast to LHCGR, the entrance to the transmembrane allosteric site of TSHR is narrower and enriched in hydrophobic amino acids. As a result, TSHR allosteric ligands, including the TP derivatives, should have hydrophobic groups that facilitate their penetration into the transmembrane channel of receptor [54,55,56]. The steric orientation of TP4/2 hydrophobic groups allows for reducing the possibility of ligand interaction with the TSHR transmembrane allosteric site. This assumption is supported by our modelling results, which show a low probability of productive protein–ligand TSHR–TP4/2 interaction, described by the GlideScore, Emodel and Lipo parameters (Table 4). This fact is confirmed by our experimental results on the absence of the TP4/2 effect on the basal and thyrotropin-releasing hormone-stimulated production of thyroid hormones (Appendix A). These data indicate the specificity of TP4/2 to the LHCGR- but not TSHR-dependent regulations, and largely exclude the negative effect of TP4/2 on the thyroid system.

The LHCGR allosteric agonists stimulate the LH/hCG signaling in target cells, including testicular Leydig cells, with less efficiency compared to gonadotropins [16,21,22,36]. This is typical for allosteric agonists, since, due to the cooperative effect, their regulatory effects quickly reach a limit with increasing concentration, and their maximal effects, as a rule, are lower than those for orthosteric agonists [57,58,59]. Nevertheless, this is rather their advantage, since it provides moderate, controlled and specific regulation of the LH/hCG pathways and, as a consequence, does not lead to hyperactivation and desensitization of LHCGR and does not induce the tissue resistance to endogenous gonadotropins. In the present study, we showed that TP4/2 is inferior to hCG in its ability to activate AC in the testicular membranes and to stimulate T production when administered to young adult male rats. In this regard, it should be noted that an increase in the intracellular level of cAMP synthesized by AC leads to activation of the StAR protein and stimulation of the steroidogenic genes expression [9,10,11,12,60,61].

We have shown for the first time that the dynamics of the steroidogenic effect of LHCGR allosteric agonist significantly differs from that of gonadotropin. With a single administration, 1–3 h after the drugs administration, the steroidogenic effect of hCG exceeded that of TP4/2, but after 5 h there was no significant difference between the steroidogenic effects of TP4/2 and hCG (Figure 2). This may be due to differences in bioavailability and pharmacokinetics of the TP derivatives and gonadotropins, their delivery route and vehicle composition [19], and to differences in the effectiveness and selectivity of their action on cAMP signaling [16,22,24]. During the five-day administration, steroidogenic effect of TP4/2 did not decrease, in contrast to that of hCG, and on the third day did not differ significantly from that of gonadotropin, as illustrated by comparable T levels in the Young+TP and Young+hCG groups (Figure 3A). It was also shown that on the fifth day of drugs treatment, there were no significant differences between the groups Young+TP and Young+hCG in the content of intratesticular T and androstenedione, although the content of 17-hydroxyprogesterone in the Young+hCG group was higher (Table 2). One of the reasons for this may be a decrease in the gene expression of LHCGR in the Young+hCG group and its significant increase in the Young+TP group (Figure 4A). It is important to note that the stimulatory effect of TP4/2 on the expression of genes encoding StAR and cytochrome CYP17A1 was not so pronounced in comparison with hCG treatment, which indicates a more moderate effect of LMW allosteric agonist on the testicular steroidogenesis (Figure 4A). Thus, five-day administration of TP4/2 provides a stable increase in T levels, but does not cause hyperactivation of the steroidogenesis system and does not suppress the LHCGR gene expression.

In women, the main area of possible use of LMW agonists of LHCGR is the assisted reproductive technologies, primarily in vitro fertilization [19,20]. In men, the prospects for their use are associated with the treatment of hypogonadotrophic hypogonadism, androgen deficiency and infertility. Production of androgens and the sperm count and quality in men are decreased in a number of metabolic and endocrine diseases, including T1DM [28,29,31], as well as in aging [32,33,34,62]. Despite the differences in etiology and pathogenesis of reproductive dysfunctions in T1DM and aging, the main causes of androgen deficiency and impaired spermatogenesis are the reduced T production by Leydig cells due to an increase in oxidative stress, inflammation and apoptosis in the testes, as well as the hormonal dysregulation due to an imbalance between signaling cascades regulated by gonadotropins, insulin, insulin-like growth factor-1 and adipokines in testicular cells [28,38,63].

We demonstrated a decrease in hCG-induced AC stimulation in the testicular membranes of diabetic and aging rats (Figure 1), as well as a weakening of the stimulating effect of hCG on the blood and intratesticular T levels and on the testicular expression of genes encoding StAR and cytochrome CYP17A1 (Figure 2, Figure 3 and Figure 4; Table 2). Since cAMP-dependent cascades are directly involved in the regulation of steroidogenesis in Leydig cells, controlling the gene expression and activity of StAR and steroidogenic enzymes [6,7,9,10,11,12], the weakening of the AC response to hCG shown by us indicates an important contribution of disturbances in the gonadotropin-regulated AC signaling system into a decrease in hCG-stimulated steroidogenesis in T1DM and aging.

In diabetic and aging rats treated with hCG, we found different changes in the testicular expression of steroidogenic genes, which we believe may be compensatory. In hCG-treated diabetic rats, expression of the *Cyp11a* gene encoding cytochrome P450_scc_, responsible for synthesis of pregnenolone, was decreased, while the expression of the *Hsd3b* gene encoding dehydrogenase HSD3β, which catalyzes the next stage of steroidogenesis, the conversion of pregnenolone into progesterone, was increased (Figure 4). At the same time, in aging rats, the stimulating effect of hCG on the *Cyp11a1* expression remained the same as in the control, and we did not detect an increase in the *Hsd3b* gene expression, as in T1DM (Figure 4). The changes in the expression of steroidogenic genes in T1DM and aging were shown by other authors [64,65,66]. Chinese scientists reported that in Leydig cells under the conditions of severe hyperglycemia and increased levels of advanced glycation end products, expression of the *Star*, *Cyp11a1* and *Hsd3b* genes was reduced [66]. In our case, the baseline expression of these genes was changed insignificantly, which may be due to a moderate form of T1DM. In aging brown Norwegian rats, along with T deficiency, a decrease in the expression of the *Star* and *Cyp11a1* genes and a decrease in the amount of StAR and cytochrome P450_scc_ proteins encoded by them were detected [64], and in two-year-old Wistar rats, the expression of the steroidogenic genes was decreased to varying degrees [65]. In our study of aging rats, there were no significant changes in the baseline expression of the steroidogenic genes, but a decrease in the hCG stimulating effect on the *Star* expression was shown (Figure 4).

In the diabetic and aging rats, as well as in young adult animals, the stimulating effect of TP4/2 on the blood T levels in the first two days of treatment was lower than for hCG, but, starting from the third day, it was similar to the hCG effect (Figure 3). The reason for this, we believe, is the maintaining of the *Lhr* gene expression in diabetic rats treated with TP4/2. In the case of hCG treatment, the *Lhr* gene expression was significantly decreased in both diabetic and aging rats, to the greatest extent in T1DM (Figure 4). It should be noted that there are many early works on a decrease in the number and expression of LHCGR in target cells when they are treated with gonadotropins with LH activity [67,68,69].

A decrease in the expression of the *Lhr* gene is due to the triggering of the mechanism of negative feedbacks, which are realized due to prolonged activation of LHCGR by gonadotropin [6]. This mechanism is due to the fact that, in contrast to effector-specific LMW allosteric agonists of LHCGR [16,22], gonadotropins activate a wide range of signaling cascades [6,7], and this can cause stimulation of intracellular pathways responsible for a decrease in the *Lhr* gene expression. Moreover, in the diabetic and aging rats, under the conditions of increased inflammation and apoptosis in the testes, the negative effect of hCG treatment on the *Lhr* expression was more pronounced than in the young adult animals. This, as can be assumed, is due to an impairment of some compensatory mechanisms that ensure the maintaining and restoration of the expression of LHCGR. Other authors showed a significant decrease in the efficiency of gonadotropin binding to LHCGR in the testes of rats with streptozotocin T1DM [66,70] and old animals [71], which is consistent with our results on the weakening of the *Lhr* expression in the diabetic and aging testes (Figure 4).

The fact that the steroidogenic effect of TP4/2 slowly but gradually increases when it is administered for five days may be due to its ability to penetrate the plasma membrane of the Leydig cells and bind to “immature” forms of LHCGR localized intracellularly, such as was demonstrated for Org 43553 by other authors [37]. Thus, the TP derivatives can function as LMW chaperones for LHCGR. This may explain the fact that in T1DM, when the post-translational processing of testicular proteins is impaired and a significant part of them are modified by glucose residues, TP4/2 retains the ability to activate LHCGR and is not inferior to hCG, despite the lower affinity for the receptor.

In the young adult, aging and diabetic rats, the stimulating effect of TP4/2 on steroidogenesis was comparable to that of hCG, but an increase in the gene expression for steroidogenesis proteins was significantly less pronounced or was not detected (Figure 4). It was shown that TP4/2′s stimulating effects on the *Star* and *Cyp17a* expression in the Young+TP group, the *Hsd3b* and *Cyp17a* expression in the Diab+TP group and the *Star* expression in the Aging+TP group were lower compared to those of hCG in the same series of animals (Figure 4). Moreover, unlike hCG, five-day treatment with TP4/2 did not affect the *Cyp11a* expression in the Young+TP, Diab+TP and Aging+TP groups and the *Hsd3b* and *Cyp17a* expression in the Aging+TP group (Figure 4). This indicates that stimulation of testicular steroidogenesis by LMW allosteric agonists does not require a significant increase in the expression of StAR and steroidogenic enzymes, which, as one can assume, is due to a more targeted activation of intracellular cascades in Leydig cells. Thus, under conditions of exposure to allosteric agonists of LHCGR, with a similar efficiency of stimulation of steroidogenesis with that of gonadotropins, a more moderate stimulation of the expression of genes of steroidogenesis occurs, without the overload of transcriptional mechanisms induced by hCG.

We demonstrated a decrease in the thickness of the seminiferous epithelium and the number of the germ and dividing cells in the seminiferous tubules of the diabetic and aging rats, which is supported by numerous experimental and clinical data of other authors [29,72,73,74,75,76,77,78]. In male patients with T1DM and in older men, along with an androgen deficiency, there are impaired spermatogenesis and sperm deterioration [29,74,75,76]. Abnormalities in the morphology of the seminiferous tubules were detected in rodents with streptozotocin diabetes and in aging animals, including thinning of the thickness of the germinal epithelium and a decrease in the number of the reproductive and dividing cells [72,73,76,77,78]. In our work, it was shown that TP4/2 restores the number of spermatogonia and spermatocytes in the seminiferous tubules of the diabetic and aging rats and increases the number of pachytene spermatocytes in young animals above the control level. In T1DM and aging, TP4/2 also restores the thickness of the seminiferous epithelium (Table 3, Figure 5, Figure 6 and Figure 7). In the control and aging rats, the effects of TP4/2 and hCG on the number of germ cells were comparable, while in the diabetic and aging animals, the effect of TP4/2 even slightly exceeded that of gonadotropin. This is the first report on the positive influence of LHCGR allosteric agonist on spermatogenesis both in young adult animals and in diabetic pathology and aging. It should be noted that hCG and other gonadotropins, including FSH, are widely used to restore spermatogenesis and fertility in male patients with hypogonadotrophic hypogonadism and other reproductive dysfunctions, but the results of such treatment are often unsuccessful, and this largely depends on the severity and origin of the impaired spermatogenesis, as well as on the etiology and pathogenesis of the reproductive disorder [79,80,81,82,83,84]. All of this suggests that TP4/2 is effective in correcting an androgen deficiency and impaired spermatogenesis caused by T1DM and aging.

## 4. Materials and Methods

### 4.1. Synthesis of 5-amino-N-tert-butyl-4-(3-(1-methylpyrazole-4-carboxamido)phenyl)-2-(methylthio)thieno[2,3-d]pyrimidine-6-carboxylic Acid Amide (TP4/2)

The synthesis of TP4/2 (Appendix A) was carried out using the acylation of 5-amino-4-(3-aminophenyl)-*N*-(*tert*-butyl)-2-(methylthio)thieno[2,3-*d*]pyrimidine-6-carboxamide, which was obtained according to the method of Hanssen and Timmers [85]. For this, 1.0 equivalent of this compound was dissolved in dry *N,N*-dimethylformamide and mixed with 1.1 equivalents of 1-methyl-*1H*-pyrazole-4-carboxylic acid in the presence of *N*,*N*-diisopropylethylamine (1.2 equivalents), and then 1.1 equivalents of 1-[bis(dimethylamino)methylene]-*1H*-1,2,3-triazolo[4,5-*b*]pyridinium 3-oxide hexafluorophosphate (HATU) were added and stirring was carried out at room temperature for 5 h. Previously, 1-methyl-*1H*-pyrazole-4-carboxylic acid was obtained via alkaline hydrolysis of the corresponding ethyl ester, which was synthesized according to the method in [86]. After removal of solvent under reduced pressure, the residue was washed with water, and the crude material was filtered off. The target product was purified by column chromatography and characterized by NMR and high-resolution mass spectrometry. The NMR spectra were obtained using a “Bruker Avance III 400” spectrometer (400.13 MHz for ^1^H and 100.61 MHz for ^13^C) (“Bruker”, Hamburg, Germany) in DMSO-*d*_6_ and were referenced to residual solvent proton signal (*δ*H = 2.50) and solvent carbon signal (*δ*C = 39.5). The high-resolution mass spectra were recorded using a “Bruker micrOTOF” spectrometer (“Bruker”, Hamburg, Germany).

The compound TP4 (C_23_H_25_N_7_O_2_S_2_) is yellow solid with the melting temperature 179–181 °C; the ^1^H-NMR spectrum (DMSO-*d*_6_), δ, ppm (*J*, Hz): 10.08 (1H, s, 3′-NH); 8.35 (1H, s, 3′’-H); 8.05 (1H, s, 5′’-H); 7.94–8.02 (2H, m, 2′-H and 6′-H); 7.56 (1H, t, *J* = 7.8, 5′-H); 7.35 (1H, d, *J* = 7.8, 4′-H); 6.98 (1H, s, NH-*t*-Bu); 6.14 (2H, s, NH_2_); 3.91 (3H, s, N-CH_3_); 2.62 (3H, s, SCH_3_); 1.37 (9H, s, C(CH_3_)_3_); and the ^13^C-NMR spectrum (DMSO-*d*_6_), δ, ppm: 168.8; 167.8; 165.2; 162.7; 161.2; 144.7; 140.0; 139.4; 137.0; 133.3; 129.7; 124.0; 122.1; 120.5; 118.7; 117.8; 97.7; 51.9 (C(CH_3_)_3_); 29.2 (C(CH_3_)_3_); 14.3 (SCH_3_). According to high-resolution mass spectrometry (ESI-TOF), the found value was 518.1422 (calculated for [M + Na^+^]: 518.1403) (Appendix A).

### 4.2. Animals

The male Wistar rats (*Rattus norvegicus*) were obtained from the “Rappolovo” animal nursery (Leningrad Region, Russia). The animals were housed in the plastic cages, five animals in each, with a 12 h light/12 h dark cycle and temperature (23 ± 3 °C) and had free access to laboratory chow pellets and water. The daily standard diet in all the studied groups contained 20–25 g of dry food (19% protein, 5% fat, 4% fiber, and 9% ash), which provided 2.95 kcal/g. Three-month-old rats were used for T1DM induction, and 18-month-old rats were considered as aging animals. The experimental procedures were approved by the Institutional Animal Care and Use Committee at the Sechenov Institute of Evolutionary Physiology and Biochemistry (St. Petersburg, Russia) (Protocol No. 01/2019, 29 January 2019) and according to “The Guide for the Care and Use of Laboratory Animals” and the European Communities Council Directive recommendations for the care and use of laboratory animals (2010/63/EU). All efforts to minimize animal suffering and to reduce the number of animals were made.

### 4.3. The Selection of the Doses and Delivery Routes for TP4/2 and hCG and the Estimation of Thyroid Hormone Levels in TP4/2-Treated Rats

The choice of the doses for TP4/2 and hCG and the method of drug delivery was carried out on the basis of preliminary studies with Young adult (four-month-old) male Wistar rats.

To determine the optimal dose for different routes of drug administration, the TP4/2 and hCG were administered intraperitoneally and subcutaneously at the doses from 5 to 50 mg/kg for TP4/2 and from 5 to 100 IU/rat for hCG. At each dose, the volume of the TP4/2 solution was 200 μL (in DMSO), and the volume of the hCG solution was 200 μL (in saline). Control animals received either DMSO (in the experiments with TP4/2) or saline (in the experiments with hCG) in the same volume and with the same route of administration. The drugs or their solvents were administered at 10.00. In these experiments, the steroidogenic effect of hCG and TP4/2 was assessed by the blood T level during 5 h after drugs administration (10.00–15.00) (Appendix A).

In the case of i.p. administration TP4/2, the maximal stimulating effect on the T production was observed at the doses 25 mg/kg and above, while in the case of s.c. administration hCG, the effect was observed at the doses 50 IU/rat and above. In accordance with this, the doses were selected in which the steroidogenic effect of TP4/2 and hCG was 70–80% of its maximum value, and these doses were 20 mg/kg for TP4/2 and 20 IU/rat for hCG in both routes of administration.

At the same time, in the case of s.c. administration, the maximum stimulating effect of TP4/2 was less pronounced and had significant variability compared to i.p. administration, which, we believe, is due to the lower bioavailability of TP4/2 when administered subcutaneously. In the case of i.p. administration of hCG, the maximum steroidogenic effect of gonadotropin was lower than in the case of its s.c. administration, and had a significant variability. Based on these data, to study the steroidogenic effects of these drugs, we used i.p. administration of TP4/2 at a dose of 20 mg/kg and s.c. administration of hCG at a dose of 20 IU/rat.

### 4.4. The Estimation of Thyroid Hormone Levels in TP4/2-Treated Rats

To assess the effect of TP4/2 on the basal and TRH-stimulated blood levels of free (fT4) and total (tT4) thyroxine and total triiodothyronine (tT3), the experiments were carried out on young adult male rats. The blood levels of thyroid hormones were measured 2 and 3.5 h after the pretreatment of the animals using i.p. injected TP4/2 in a single dose of 20 mg/kg. TRH (300 μg/kg, 20 μL) or saline (20 μL) was administered intranasally 30 min after pretreatment with TP4/2. The details of the experiment are given in the caption to Appendix A.

### 4.5. Young Adult Rats and Their Treatment

Young adult (four-month-old) male Wistar rats were randomized into four groups, which for five days were treated with DMSO (i.p.) (Young+dmso, *n* = 5), TP4/2 (i.p., in DMSO) at a daily dose of 20 mg/kg (Young+TP, *n* = 5), saline (s.c.) (Young+saline, *n* = 5), or hCG (s.c., in saline) at a daily dose of 20 IU/rat (Young+hCG, *n* = 5) (Appendix A). The drugs or their solvents were administered at 10.00. The volume of the hCG solution was 200 μL (in saline), and the volume of the TP4/2 solution was 200 μL (in DMSO). The DMSO (i.p.) and saline (s.c.) were injected in the same volumes. On the fifth day, 3 h after treatment, the rats were sacrificed under anesthesia, which was carried out by inhalation of 4–5% (v/v) isoflurane. The testes were dissected immediately and frozen in liquid nitrogen for the RT-PCR and ELISA or for the preparation of testicular membranes. For histochemical analysis, the testes were fixed in 4% *para*-formaldehyde solution. Based on the body and testes weight, the gonadosomatic index (GSI) was calculated as (testes weight/total body weight) × 100. The body weight, testes mass, metabolic and hormonal parameters in young adult rats are given in Table 1.

### 4.6. Aging Rats

For the study, intact 18-month-old male Wistar rats were selected, which, like young adult animals, were divided into four groups treated with DMSO (i.p.) (Aging+dmso, *n* = 5), TP4/2 (i.p., 20 mg/kg/day) (Aging+TP, *n* = 5), saline (s.c.) (Aging+saline, *n* = 5), or hCG (s.c., 20 IU/rat/day) (Aging+hCG, *n* = 5) (see Appendix A). The body weight, testes mass, metabolic and hormonal parameters in aging rats are given in Table 1.

### 4.7. Diabetic Rats and Their Treatment

To induce T1DM, three-month-old male Wistar rats were injected with streptozotocin (i.p., 50 mg/kg). The development of T1DM was estimated 10 days after streptozotocin injection according to postprandial hyperglycemia (2 h after food intake). For further experiments, the rats with the glucose levels >15 mmol/L were selected. Glucose concentration in the whole blood from the tail vein was measured using a glucometer and the test strips “One Touch Ultra” (“LifeScan Inc.”, Malvern, Pennsylvania, USA). On the 30th day after T1DM induction, the diabetic rats were randomized into four groups, which for five days were treated with DMSO (i.p.) (Diab+dmso, *n* = 5), TP4/2 (i.p., 20 mg/kg/day) (Diab+TP, *n* = 5), saline (s.c.) (Diab+saline, *n* = 5), or hCG (s.c., 20 IU/rat/day) (Diab+hCG, *n* = 5) (see Appendix A). The body weight, testes mass, metabolic and hormonal parameters in diabetic rats are given in Table 1.

### 4.8. Chemicals and Radiochemicals

Streptozotocin, hCG, thyroid stimulating hormone (TSH) from bovine pituitary, thyrotropin-releasing hormone (TRH), forskolin, 5′-guanylylimidodiphosphate (GppNHp), creatine phosphate, creatine phosphokinase from the rabbit muscle, sodium glycerophosphate, sodium deoxycholate, phenylmethylsulfonyl fluoride, *ortho*-phenanthroline, pepstatin and other reagents were obtained from “Sigma-Aldrich” (St. Louis, MO, USA). Radiolabeled substrate for AC, [α-^32^P]-ATP was obtained from “Izotope” (Moscow, Russia).

### 4.9. The Determination of the Blood Levels of Testosterone, Luteinizing Hormone, Insulin, Leptin, Glycated Hemoglobin (HbA1c) and Thyroid Hormones

The blood levels of testosterone (T) were estimated with the “Testosterone-ELISA kit” (“Alkor-Bio”, Saint Petersburg, Russia). The HbA1c levels were measured using the “Multi Test HbA1c System kit” (“Polymer Technology Systems, Inc.”, Indianapolis, IN, USA). The serum levels of insulin, leptin and LH were measured with the “Rat Insulin ELISA kit” (“Mercodia AB”, Uppsala, Sweden), “ELISA for Leptin, Rat” and “ELISA for LH, Rat” (“Cloud-Clone Corp.”, Houston, TX, USA), respectively. The blood levels of free (fT4) and total (tT4) thyroxine and total triiodothyronine (tT3) were determined using the ELISA kits obtained from “Immunotech” (Moscow, Russia).

### 4.10. The Determination of the Intratesticular Levels of Testosterone and Its Precursors

To determine the intratesticular content of sex steroid hormones, the testes samples were homogenized (1:10, *v*/*v*) in the lysis buffer containing 20 mM Tris-HCl (pH 7.5), 150 mM NaCl, 2 mM EDTA, 2 mM EGTA, 0.25 M sucrose, 0.5% Triton X-100, 0.5% sodium deoxycholate, 15 mM NaF, 10 mM sodium glycerophosphate, 10 mM sodium pyrophosphate, 1 mM sodium *ortho*-vanadate, 1 mM phenylmethylsulfonyl fluoride, 500 µM *ortho*-phenanthroline, 2 μM pepstatin, and 0.02% sodium azide, as described earlier [87]. The obtained homogenate was centrifuged (10,000× *g*, 5 min), and the content of steroid hormones in supernatant was measured according to the manufacturer’s instructions. The intratesticular content of T, progesterone and 17-hydroxyprogesterone were estimated with the “Testosterone-ELISA kit” (“Alkor-Bio”, Saint Petersburg, Russia), “Progesterone-EIA” and “17-OH-progesterone-EIA” (“XEMA Co. Ltd.”, Moscow, Russia), respectively. The intratesticular content of androstenedione was determined using the “Androstenedione ELISA” (“DRG Instruments GmbH”, Marburg, Germany).

### 4.11. The Isolation of Testicular and Thyroidal Membranes

The isolation of plasma membrane fractions from the testes and thyroid gland of rats was carried out as described previously [88,89]. The testes tissue was obtained from decapitated rats and was crushed and placed in ice-cold 40 mM Tris-HCl buffer (pH 7.4) containing 5 mM MgCl_2_, 320 mM sucrose and a cocktail of protease inhibitors (500 μM *ortho*-fenanthroline, 2 µM pepstatin, and 1 mM phenylmethylsulfonyl fluoride). The testicular tissue was homogenized with a Polytron, and the homogenate was centrifuged at 1500× *g* for 10 min at 4 °C. The supernatant was centrifuged at 20,000× *g* for 30 min at 4 °C. The resulting pellet was washed by resuspension in 10 volumes of the homogenizing buffer (without sucrose) and centrifuged at 20,000× *g* for 30 min at 4 °C. The thyroid tissue was washed with ice-cold 40 mM Tris-HCl buffer (pH 7.4) containing 5 mM MgCl_2_ and a cocktail of protease inhibitors, and then cut into small pieces and homogenized with the same buffer. Further, the homogenate was centrifuged at 500× *g* for 15 min at 4 °C. The resulting pellet was discarded, and the supernatant centrifuged at 10,000× *g* for 30 min at 4 °C. The final pellet obtained from the rat testes and thyroid gland was resuspended in 50 mM Tris-HCl buffer (pH 7.4) to produce the membrane fraction with a protein concentration range of 1–3 mg/mL and stored at −80 °C. The protein concentration of each membrane preparation was measured using the method of Lowry and colleagues using BSA as a standard.

### 4.12. Adenylyl Cyclase Assay

The adenylyl cyclase (EC 4.6.1.1) activity was measured using the method of Salomon and coauthors [90], with some modifications [91]. The reaction mixture (final volume 50 μL) contained 50 mM Tris-HCl (pH 7.5), 5 mM MgCl_2_, 1 mM ATP, 1 μCi [α-^32^P]-ATP, 0.1 mM cAMP, 20 mM creatine phosphate, 0.2 mg/mL creatine phosphokinase, and 40–80 μg of the membrane protein. Incubation was carried out at 37 °C for 12 min. The reaction was initiated by the addition of membrane fraction and stopped by the addition of 100 μL of 0.5 M HCl, followed by immersing the tubes with the mixture in a boiling water bath for 6 min. Then, 100 μL of 1.5 M imidazole was added to each tube for the neutralization of HCl. The [^32^P]-cAMP formed as a result of the enzymatic reaction was separated using the column chromatography. The samples were placed on neutral alumina columns and cAMP was eluted with 12 mL of 10 mM imidazole-HCl buffer (pH 7.4). The eluates were collected in scintillation vials and counted using a 1209/1215 RackBeta scintillation counter (“LKB”, Stockholm, Sweden). Each assay was carried out in triplicate at least three times, and the results were expressed as pmol cAMP/min/mg of membrane protein. The basal activity was measured in the absence of hormonal agents, GppNHp and forskolin.

### 4.13. The RNA Extraction, Reverse Transcription and Real-Time Quantitative PCR Analysis

Total RNA was isolated from the testes using the ExtractRNA Reagent (TRIzol analogue) (“Evrogen”, Moscow, Russia) according to the manufacturer’s protocol. Before use, the RNA samples were examined by agarose gel electrophoresis to demonstrate clear bands corresponding to the ribosomal 5S/5.8S, 18S and 28S RNA and no degradation. The quality and yield of the RNA were estimated by measuring the A260/A280 ratio (Nanophotometer C40, “Implen”, Munchen, Germany). The samples containing 1 μg of RNA were reverse-transcribed to cDNA using the random oligodeoxynucleotide primers and the MMLV RT kit (“Evrogen”, Moscow, Russia) according to the manufacturer‘s protocol. The amplification was performed using the mixture (final volume of 25 μL) containing 10 ng of the reverse transcription product, 0.4 μM each of the forward and reverse primers, and the qPCRmix-HS SYBR+LowROX kit (“Evrogen”, Moscow, Russia), as described by us earlier [92]. The amplified signals were detected with the Applied Biosystems^®^ 7500 Real-Time PCR System (“Life Technologies, Thermo Fisher Scientific Inc.”, Waltham, MA, USA). The real-time quantitative PCR amplification protocol was used: (i) initial denaturation at 95 °C for 5 min; (ii) three-stage amplification and quantification program consisting of 38 cycles of 95 °C for 30 s, 55–58 °C for 10 s, and 72 °C for 30 s; and (iii) use of the ABI Melt Curve program to verify the presence of a single peak and the absence of primer-dimer formation in each template-containing reaction. The annealing temperatures were optimized using the Primer-Blast program (http://www.ncbi.nlm.nih.gov/tools/primer-blast/). In the preliminary studies, the SYBR Green-labeled PCR products were evaluated by agarose gel electrophoresis, and the authenticity of each amplicon was verified by nucleic acid sequencing. The primers for the investigated and reference genes listed in Appendix A. The expression of the genes encoding β-actin (*Actb*) was used as a house kipping gene. The data were calculated using the delta-delta C_t_ method and expressed as fold expression relative to the control [93]. In all the studied series, young adult rats were used as a control group. Since there were no differences in gene expression in the groups Young+saline and Young+dmso, the group Young+saline was used as a control to assess the expression of testicular genes.

### 4.14. Preparation of the Testis Sections

The testes were fixed for 48 h (4 °C) in 4% *p*-formaldehyde solution (“Sigma-Aldrich”, St. Louis, MO, USA), washed by 0.9% Na-phosphate buffer (PBS) and immersed in PBS containing 30% sucrose at +4 °C. The testes were frozen on dry ice using Tissue-Tek^®^ medium (“Sacura Finetek Europe”, Alphen aan den Rijn, The Netherlands). The series of cross sections from different levels of the testes (6 μm) were prepared using a Leica CM-1520 cryostat (“Leica Microsystems”, Nussloch, Germany) and mounted on the SuperFrost/plus glasses (“Menzel”, Braunschweig, Germany). Sections from the investigated groups were placed on the same glass, dried overnight and used for histochemical analysis.

### 4.15. Histological Analysis of the Seminiferous Tubules

For the study of the morphology of the seminiferous tubules, the testis sections were washed with 50% ethanol for 15 min, then distilled water and stained with Ehrlich hematoxylin and Eosin G according to the standard procedure and placed under a cover glass using glycerol [94,95]. The samples were analyzed using a Carl Zeiss Imager A1 microscope (“Carl Zeiss AG”, Oberkochen, Germany). Using a ×40 objective on the sections obtained from different levels of the testis, the seminiferous tubules were selected, and the number of spermatogonia and pachytene spermatocytes was counted on the same area. Thus, 10 tubules were analyzed in each animal, and then the mean of the parameters evaluated was calculated. To estimate the parameters, two independent investigators performed measurements. Each researcher who carried out the evaluation did not initially know the identity of any of the specimens with respect to treatment and group assignment. Using an ×20 objective and Carl Zeiss software (Axio Vision 4.7.2, “Carl Zeiss AG”, Oberkochen, Germany), the microphotography was made and the thickness of the seminiferous epithelium (µm) was measured, estimated as the distance from the basal lamina to the head of mature spermatozoa. Ten tubules were counted for each animal, and two measurements for each tubule were made [95].

### 4.16. Modelling of the Rat LHCGR Structure and Thyrotropin-Stimulating Hormone (TSH) Receptor (TSHR)

The structures of the rat LHCGR and TSHR modeled with the use of homology modeling methods and molecular dynamics (Appendix A). The template structures for receptor modeling were taken from RCSB Protein Data bank and GPCR db. The initial three-dimensional structure of transmembrane domain of the rat LHCGR and TSHR was built on the basis of the 3D models of bovine rhodopsin (Protein Data Bank ID: 1GZM, 4WW3, 1L9H) [96]. Several receptor-specific corrections were made based on sequence alignments using Schrodinger Suite 2020-2. The transmembrane domain design was based on the Ballesteros-Weinstein nomenclature. The secondary structure of the transmembrane domains was built in accordance with the models of the human LHCGR and TSHR, from the GPCR database [97]. Amino acid sequence mismatches between proteins were fixed by the homology structure modeling with use of multiple sequence alignment. Missing loops were added by best fit and homology to fragments of other proteins (with Prime module from Schrodinger Suite 2020-2) [98]. The stability of the protein structure was studied by molecular dynamics runs in a water (TIP3P, simulated by 0.15 M NaCl solution) box (20 Å buffer zone from protein) system without any restraints (15 ns, periodic boundary box, charges neutralized by adding chlorine ions) using Desmond software [99]. The stability and quality of the model were controlled by checking the geometry with Ramachandran plot and during the MD simulation (backbone RMSD less than 2 Å). All MD manipulations processed in OPLS3e force field [100].

### 4.17. Docking of Thienopyrimidine Derivatives and Parameters Selected for Analysis

For each structure, the grid box was built in accordance with coordinates of the allosteric ligand-binding sites, as described by other authors [101,102]. The grid dimension is a 12 Å buffer zone in accordance to reference ligand size. Docking was made without any constraints. All manipulations were processed in OPLS3e force field [100]. The studied compounds, TP4/2 and Org43553, were prepared with LigPrep module from Schrodinger Suite 2020-2. For each compound, 2000 docking solutions were generated, and twenty of them were selected for analysis. Best solutions were selected in accordance with its cluster volume. All docking procedures were carried out with Glide module from Schrodinger Suite 2020-2 [103].

For the analysis, the parameters such as GlideScore, Emodel and Lipo/Coulomb/H-bond were selected. GlideScore is an empirical scoring function that approximates the ligand binding free energy. It has many terms, including force field (electrostatic, van der Waals) contributions and terms rewarding or penalizing interactions known to influence ligand binding. It has been optimized for docking accuracy, database enrichment, and binding affinity prediction. GlideScore should be used to rank poses of different ligands. As it simulates binding free energy, more negative values represent better binding affinity. The Emodel parameter has a more significant weighting of the force field components (electrostatic and van der Waals energies), which makes it well-suited for comparing conformers, but much less so for comparing chemically-distinct species. Therefore, the Glide program uses the Emodel value to pick the “best” pose of a ligand (pose selection), and then ranks these best poses against one another with GlideScore. Lipo/Coulomb/H-bonds parameters are basic binding energy components which can be used for ligand affinity differentiation by preferred interaction type and their increment, such as the hydrophobic, electrostatic or hydrogen bonding.

### 4.18. Statistical Analysis

All experimental data were analyzed using the software IBM SPSS Statistics 23.0.0.0 (“IBM”, New York, NY, USA). The multiple comparisons between the groups of rats was assessed statistically using the one-way analysis of variance (ANOVA) and multivariate general linear model analysis with the Bonferroni post hoc test, and results are presented as the mean ± standard error of the mean (M ± SEM). All differences are considered as significant at *p <* 0.05. The EC_50_ value for the “dose-effect” curve was estimated with GraphPad Prism 8.4.3, using nonlinear regression analysis with calculating of four parameter variable slope.

## 5. Conclusions

We have shown that TP4/2, a LHCGR allosteric agonist developed by us, stimulates T production in male Wistar rats during a five-day administration. With a single administration, TP4/2 is inferior in the steroidogenic effect to hCG, which is widely used in the clinic to stimulate steroidogenesis, but with a long-term use, on the third to fifth days after administration, the steroidogenic effect of TP4/2 is comparable to that of gonadotropin. The compound TP4/2 is effective not only in the treatment of young adult rats, but also in the treatment of both aging, 18-month-old, rats and diabetic rats with streptozotocin-induced T1DM. When administered to the aging and diabetic rats, TP4/2 is as effective as hCG in normalizing T levels and improving testicular morphology and germ cell maturation. Unlike hCG, TP4/2 does not cause a decrease in the *Lhr* gene expression, thereby preventing the resistance of Leydig cells to endogenous gonadotropins, and also does not cause hyperactivation of the gene expression of cholesterol-transporting StAR protein and some steroidogenic enzymes. A more moderate, but at the same time more stable stimulation of steroidogenesis, induced by TP4/2, demonstrates good prospects for its use of TP4/2 and, as one can assume, other LMW allosteric full agonists of LHCGR to compensate for androgen deficiency, both in certain physiological states, including aging, starvation and others, as well as in a number of diseases, which are characterized by an impairment of the hypothalamic–pituitary–gonad axis, including diabetes mellitus and metabolic syndrome.

New results obtained in this study, as well as earlier data on the study of other TP derivatives [23] show that, unlike gonadotropins, the LMW allosteric agonists of LHCGR can be used for a long time, but not cause hyperactivation of both LH signaling system and steroidogenesis in Leydig cells. In addition, taking into account chaperone-like properties of TP derivatives in relation to LHCGR [37] and the absence of a decrease in the *Lhr* gene expression in the case of long-term TP4/2 treatment demonstrated in our study, it can be assumed that the LMW allosteric agonists are able to enhance the effects of LH and hCG, and this will reduce the dose of clinically used gonadotropins, thus preventing some of their side effects.

## Figures and Tables

**Figure 1 ijms-21-07493-f001:**
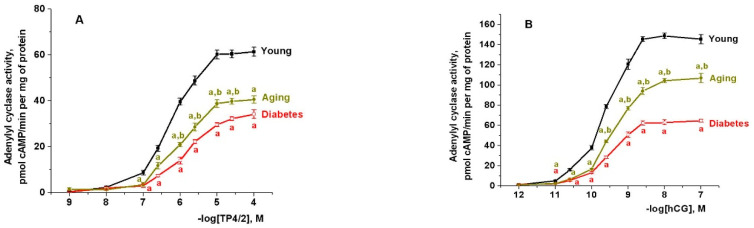
The concentration-dependent stimulating effects of TP4/2 (**A**) and hCG (**B**) on the basal adenylyl cyclase activity in testicular membranes isolated from the young adult, aging and diabetic rats. ^a^—the difference between the young adult rats and the aging and diabetic rats is significant at *p* < 0.05; ^b^—the difference between the aging and diabetic rats is significant at *p* < 0.05. The data are presented as the M ± SEM, *n* = 5.

**Figure 2 ijms-21-07493-f002:**
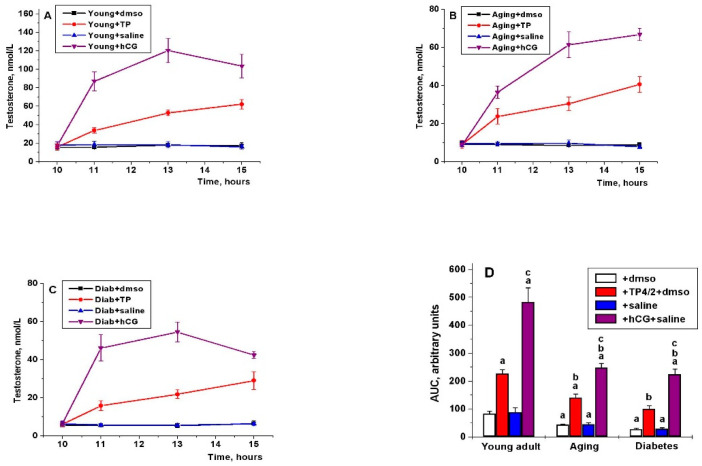
The stimulating effects of a single administration of TP4/2 and hCG on the blood testosterone levels in the young adult, aging and diabetic male rats. (**A**) Young+dmso—young adult rats plus DMSO, i.p.; Young+TP—young adult rats plus TP4/2, 20 mg/kg, i.p.; Young+saline—young adult rats plus saline, s.c.; Young+hCG—young adult rats plus hCG, 20 IU/rat, s.c.; (**B**) Aging+dmso—aging rats plus DMSO, i.p.; Aging+TP—aging rats plus TP4/2, 20 mg/kg, i.p.; Aging+saline—aging rats plus saline, s.c.; Aging+hCG—aging rats plus hCG, 20 IU/rat, s.c.; (**C**) Diab+dmso—diabetic rats plus DMSO, i.p.; Diab+TP—diabetic rats plus TP4/2, 20 mg/kg, i.p.; Diab+saline—diabetic rats plus saline, s.c.; Diab+hCG—diabetic rats plus hCG, 20 IU/rat, s.c.; (**D**) the AUC_10.00–15.00_ value for the “T concentration (nmol/L)–time (hours)” curves calculated for each of the studied groups and were given in relative units. The drugs were administered at 10.00. ^a^—the difference between the group Young+dmso and the other groups with the treatment DMSO or TP4/2+DMSO or between the group Young+saline and the other groups treated with saline or hCG+saline is significant at *p* < 0.05; ^b^—the difference between aging/diabetic rats without TP4/2 or hCG administration and aging/diabetic rats with TP4/2 or hCG administration is significant at *p* < 0.05; ^c^—the difference between the TP4/2- and hCG-treated rats within the young adult, aging and diabetic series is significant at *p* < 0.05. The data are presented as the M ± SEM, *n* = 5.

**Figure 3 ijms-21-07493-f003:**
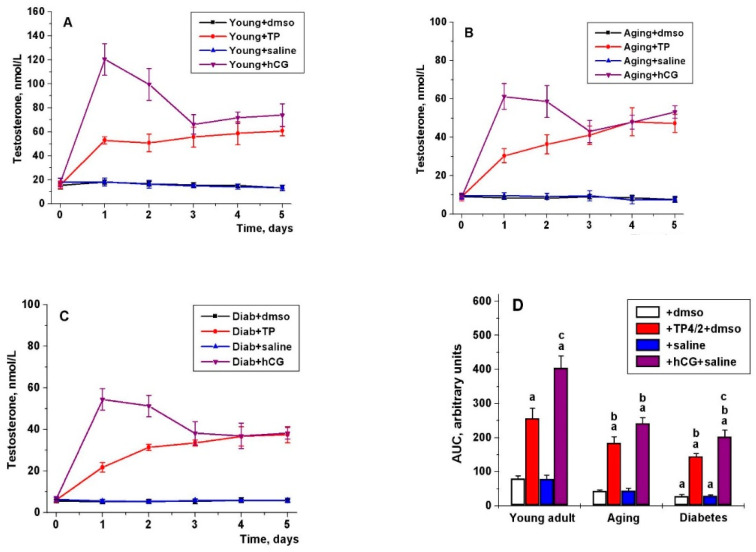
The effects of five-day treatment of the young adult, aging and diabetic male rats with TP4/2 and hCG on the blood testosterone levels. (**A**) Young+dmso—young adult rats plus DMSO, i.p.; Young+TP—young adult rats plus TP4/2, 20 mg/kg, i.p.; Young+saline—young adult rats plus saline, s.c.; Young+hCG—young adult rats plus hCG, 20 IU/rat, s.c.; (**B**) Aging+dmso—aging rats plus DMSO, i.p.; Aging+TP—aging rats plus TP4/2, 20 mg/kg, i.p.; Aging+saline—aging rats plus saline, s.c.; Aging+hCG—aging rats plus hCG, 20 IU/rat, s.c.; (**C**) Diab+dmso—diabetic rats plus DMSO, i.p.; Diab+TP—diabetic rats plus TP4/2, 20 mg/kg, i.p.; Diab+saline—diabetic rats plus saline, s.c.; Diab+hCG—diabetic rats plus hCG, 20 IU/rat, s.c.; (**D**) the AUC_0–5_ value for the “T concentration (nmol/L)–time (days)” curves calculated for each of the studied groups and were given in relative units; (**E**) the ratio between stimulating effects of TP4/2 and hCG (the T concentration increases caused by these drugs over the baseline T concentration at the day “0”). The drugs were administered every day at 10.00, and the blood T levels were measured 3 h after the treatment (at 13.00), when the steroidogenic effects reaches saturation. ^a^—the difference between the group Young+dmso and the other groups with the treatment DMSO or TP4/2+DMSO or between the group Young+saline and the other groups treated with saline or hCG+saline is significant at *p* < 0.05; ^b^—the difference between aging/diabetic rats without TP4/2 or hCG administration and aging/diabetic rats with TP4/2 or hCG administration is significant at *p* < 0.05; ^c^—the difference between the TP4/2- and hCG-treated rats within the young adult, aging and diabetic series is significant at *p* < 0.05. The data are presented as the M ± SEM, *n* = 5.

**Figure 4 ijms-21-07493-f004:**
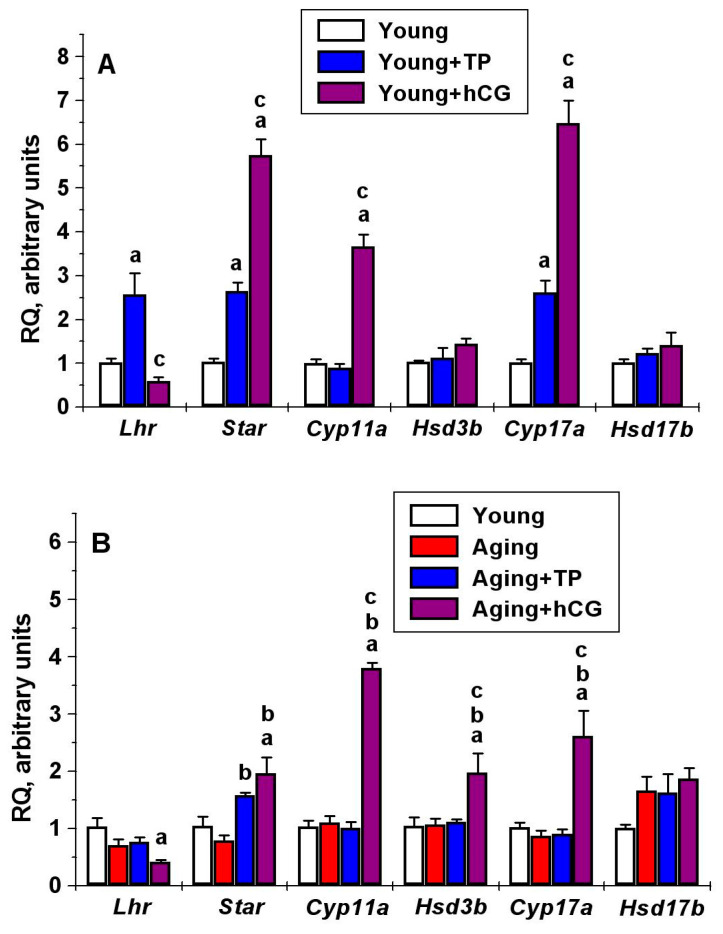
The gene expression in the testes of the young adult (**A**), aging (**B**) and diabetic (**C**) male rats, and the effect of five-day treatment with TP4/2 and hCG. ^a^—the difference between the young adult rats without TP4/2 or hCG treatment and the other groups is significant at *p* < 0.05; ^b^—the difference between aging/diabetic rats without TP4/2 or hCG administration and aging/diabetic rats with TP4/2 or hCG administration is significant at *p* < 0.05; ^c^—the difference between the TP4/2- and hCG-treated rats within the young adult, aging and diabetic series is significant at *p* < 0.05. TP4/2 was administered intraperitoneally at a daily dose of 20 mg/kg, and hCG was administered subcutaneously at a daily dose of 20 IU/rat. The data are presented as the M ± SEM, *n* = 5.

**Figure 5 ijms-21-07493-f005:**
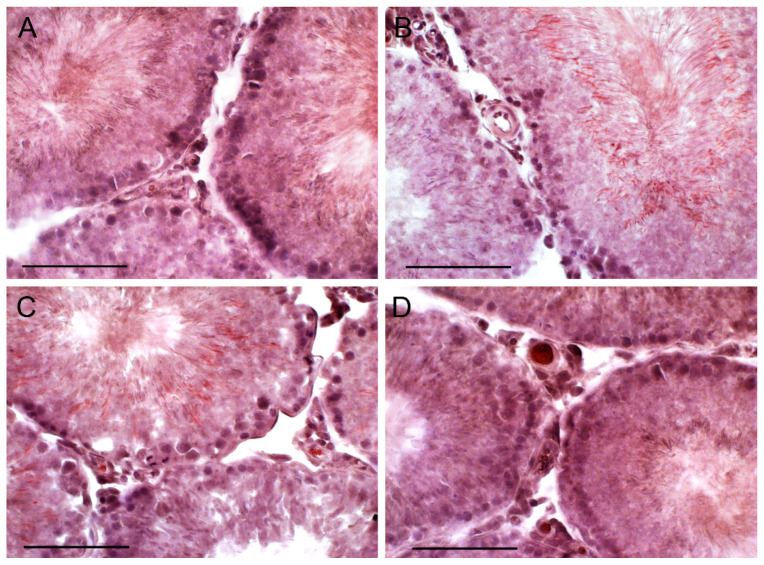
The morphology of the seminiferous tubules in the testis of young adult rats, and the effect of five-day treatment with TP4/2 and hCG. **A**—Young+dmso, **B**—Young+TP, **C**—Young+saline; **D**—Young+hCG. The histology of the rat testes was evaluated using the staining of the testis section with Ehrlich hematoxylin and Eosin G. Scale bars, 100 µm.

**Figure 6 ijms-21-07493-f006:**
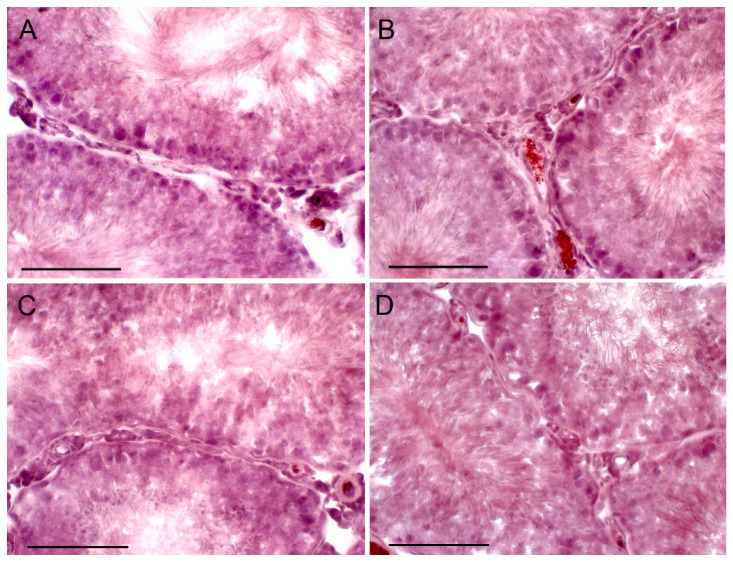
The morphology of the seminiferous tubules in the testis of aging rats, and the effect of a five-day treatment with TP4/2 and hCG. **A**—Aging+dmso, **B**—Aging+TP, **C**—Aging+saline, **D**—Aging+hCG. The histology of the rat testis was evaluated using the staining of the testis section with Ehrlich hematoxylin and Eosin G. Scale bars, 100 µm.

**Figure 7 ijms-21-07493-f007:**
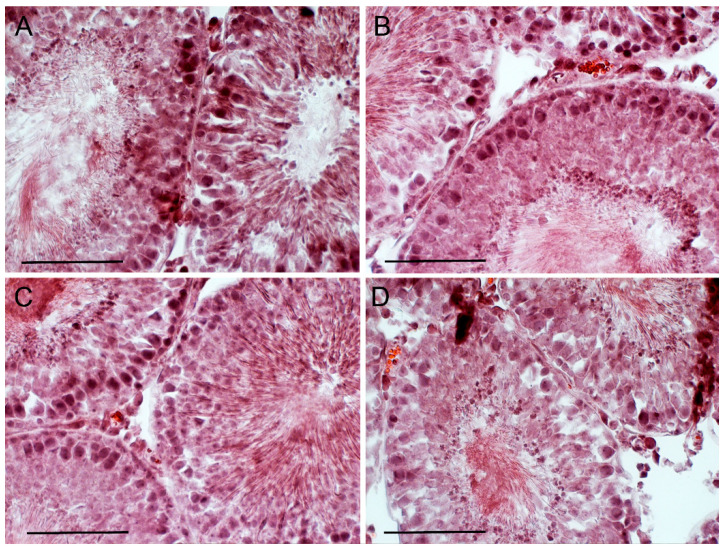
The morphology of the seminiferous tubules in the testis of diabetic rats, and the effect of five-day treatment with TP4/2 and hCG. **A**—Diab+dmso, **B**—Diab+TP, **C**—Diab-saline, **D**—Diab+hCG. The histology of the rat testes was evaluated using the staining of the testis section with Ehrlich hematoxylin and Eosin G. Scale bars, 100 µm.

**Figure 8 ijms-21-07493-f008:**
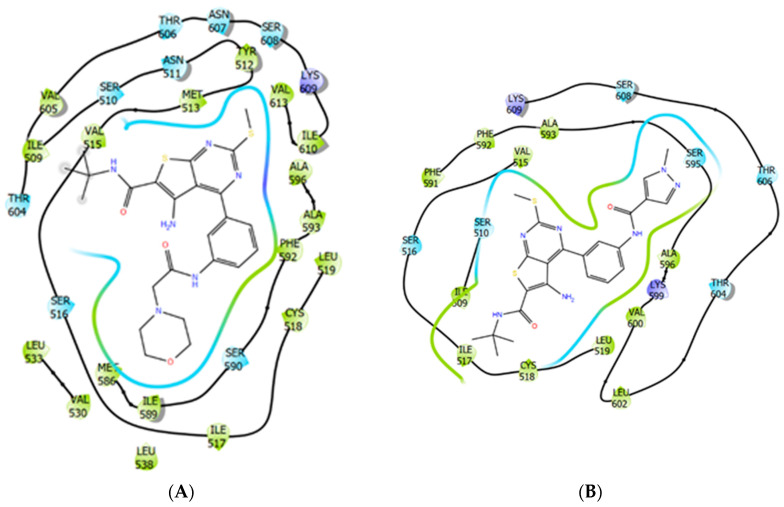
The molecular docking of the thienopyrimidine derivatives Org 43553 (reference compound) (**A**) and TP4/2 (testing compound) (**B**) into the allosteric site of rat LHCGR. Amino acid residues with the hydrophobic (in green), positively charged (in violet) or polar groups (in blue) are shown. The same colors are used to indicate contacts between the allosteric agonist and the surface of the LHCGR allosteric site.

**Table 1 ijms-21-07493-t001:** The body and testes weight, gonadosomatic index (GSI), and blood levels of glucose, glycated hemoglobin (HbA1c), insulin, leptin and luteinizing hormone (LH) in young adult, aging and diabetic male Wistar rats.

Groups	Body Weight, g	Testes Weight, g	GSI, arb. Units	Glucose, mmol/L ^@^	HbA1c, %	Insulin, ng/mL #	Leptin, ng/mL #	LH, pg/mL #
Young+dmso	315 ± 8	3.84 ± 0.15	1.22 ± 0.03	5.2 ± 0.2	4.10 ± 0.11	0.70 ± 0.16	2.45 ± 0.35	1.97 ± 0.27
Young+TP	320 ± 5	4.02 ± 0.11	1.26 ± 0.02	5.3 ± 0.2	4.20 ± 0.13	0.76 ± 0.14	2.34 ± 0.40	1.30 ± 0.17
Young+saline	329 ± 7	3.90 ± 0.05	1.19 ± 0.02	5.4 ± 0.1	4.24 ± 0.12	0.80 ± 0.14	2.60 ± 0.32	1.94 ± 0.24
Young+hCG	322 ± 8	4.34 ± 0.14 ^a^	1.35 ± 0.03 ^a^	5.5 ± 0.2	4.12 ± 0.09	0.98 ± 0.15	2.64 ± 0.25	0.77 ± 0.11 ^a^
Aging+dmso	451 ± 14	4.14 ± 0.19	0.92 ± 0.03	6.4 ± 0.4	4.78 ± 0.29	1.07 ± 0.15	3.23 ± 0.33	2.11 ± 0.27
Aging+TP	454 ± 9 ^a^	4.36 ± 0.13	0.96 ± 0.02 ^a^	6.1 ± 0.3	4.48 ± 0.25	0.99 ± 0.16	3.13 ± 0.42	1.76 ± 0.23
Aging+saline	458 ± 13 ^a^	4.16 ± 0.10	0.91 ± 0.05 ^a^	6.2 ± 0.3	4.68 ± 0.22	1.14 ± 0.20	3.38 ± 0.41	2.37 ± 0.32
Aging+hGC	450 ± 6 ^a^	4.80 ± 0.18 ^a,b^	1.07 ± 0.04 ^a^	6.4 ± 0.3	4.46 ± 0.29	1.08 ± 0.17	2.92 ± 0.26	0.83 ± 0.09 ^a,b^
Diab+dmso	284 ± 6	3.26 ± 0.07	1.15 ± 0.01	22.3 ± 2.1	7.36 ± 0.59	0.14 ± 0.04	1.39 ± 0.22	1.16 ± 0.19
Diab+TP	283 ± 7 ^a^	3.44 ± 0.07 ^a^	1.22 ± 0.02	20.4 ± 1.8 ^a^	6.66 ± 0.20 ^a^	0.20 ± 0.05 ^a^	1.55 ± 0.21 ^a^	1.01 ± 0.20 ^a^
Diab+saline	292 ± 8	3.30 ± 0.10 ^a^	1.13 ± 0.04	21.8 ± 1.6 ^a^	7.10 ± 0.42 ^a^	0.17 ± 0.06 ^a^	1.49 ± 0.22 ^a^	1.22 ± 0.22
Diab+hCG	290 ± 7 ^a^	3.66 ± 0.11	1.26 ± 0.01 ^b^	19.1 ± 1.9 ^a^	6.44 ± 0.39 ^a^	0.16 ± 0.04 ^a^	1.32 ± 0.28 ^a^	0.72 ± 0.14 ^a^

Notes: @—the postprandial glucose levels in young adult, aging and diabetic rats, 2 h after food intake. #—the blood insulin, leptin and LH levels were estimated in fasted animals (12 h of fasting). ^a^—the difference between the group Young+dmso and the other groups with the treatment DMSO or TP4/2+DMSO or between the group Young+saline and the other groups treated with saline or hCG+saline is significant at *p* < 0.05; ^b^—the difference between aging/diabetic rats without TP4/2 or hCG administration and aging/diabetic rats with TP4/2 or hCG administration is significant at *p* < 0.05. In each group *n* = 5, M ± SEM.

**Table 2 ijms-21-07493-t002:** The intratesticular content of testosterone and its precursors in the young adult, aging and diabetic rats, and the effect of five-day treatment with TP4/2 and hCG.

Group	Progesterone, nmol/g of Tissue	17-OH-Progesterone, nmol/g of Tissue	Androstenedione, pmol/g of Tissue	Testosterone, nmol/g of Tissue
Young+dmso	0.536 ± 0.036	0.063 ± 0.004	53.8 ± 4.2	0.995 ± 0.105
Young+TP	0.534 ± 0.088	0.173 ± 0.012	148.6 ± 10.1 ^a^	2.174 ± 0.244 ^a^
Young+saline	0.569 ± 0.041	0.066 ± 0.005	55.6 ± 6.7	0.910 ± 0.086
Young+hCG	0.545 ± 0.066	0.395 ± 0.055 ^a,c^	214.3 ± 11.0 ^a,c^	2.810 ± 0.132 ^a^
Aging+dmso	0.728 ± 0.064 ^a^	0.025 ± 0.005 ^a^	32.1 ± 5.6	0.255 ± 0.044 ^a^
Aging+TP	0.856 ± 0.043 ^a^	0.075 ± 0.005	129.1 ± 6.7 ^a,b^	1.441 ± 0.078 ^a,b^
Aging+saline	0.699 ± 0.034	0.019 ± 0.003 ^a^	30.4 ± 3.0	0.237 ± 0.043 ^a^
Aging+hGC	1.558 ± 0.164 ^a,b,c^	0.271 ± 0.022 ^a,b,c^	233.9 ± 16.5 ^a,b,c^	1.869 ± 0.123 ^a,b,c^
Diab+dmso	0.377 ± 0.030	0.049 ± 0.006	32.7± 5.1	0.287 ± 0.032 ^a^
Diab+TP	0.521 ± 0.048	0.083 ± 0.006^a^	100.3 ± 16.4 ^a,b^	1.308 ± 0.097 ^a,b^
Diab+saline	0.387 ± 0.026	0.057 ± 0.008	35.1± 5.6	0.293 ± 0.044 ^a^
Diab+hGC	0.480 ± 0.087	0.142 ± 0.011 ^a,b,c^	141.5 ± 10.0 ^a,b^	1.356 ± 0.117 ^a,b^

Notes: ^a^—the difference between the group Young+dmso and the other groups with the treatment DMSO or TP4/2+DMSO or between the group Young+saline and the other groups treated with saline or hCG+saline is significant at *p* < 0.05; ^b^—the difference between aging/diabetic rats without TP4/2 or hCG administration and aging/diabetic rats with TP4/2 or hCG administration is significant at *p* < 0.05; ^c^—the difference between the TP4/2- and hCG-treated rats within the young adult, aging and diabetic series is significant at *p* < 0.05. TP4/2 was administered intraperitoneally at a daily dose of 20 mg/kg, and hCG was administered subcutaneously at a daily dose of 20 IU/rat. The data are presented as the M ± SEM, *n* = 5.

**Table 3 ijms-21-07493-t003:** The effect of five-day treatment of young adult, aging and diabetic rats with TP4/2 and hCG on the thickness of the seminiferous epithelium and the number of germ cells in the seminiferous tubules.

Group	Thickness of the Seminiferous Epithelium, µm	Number of Spermatogonia/Seminiferous Tubule, Units	Number of Pachytenic Spermatocytes/Seminiferous Tubule, Units
Young+dmso	82.64 ± 0.94	59.18 ± 0.77	59.4 ± 1.48
Young+TP	81.91 ± 0.68	60.60 ± 0.96	68.54 ± 1.20 ^a^
Young+saline	83.21 ± 0.63	58.78 ± 0.85	60.32 ± 1.33
Young+hCG	81.15 ± 0.87	64.00 ± 1.02 ^a^	73.60 ± 1.34 ^a^
Aging+dmso	66.95 ± 0.88 ^a^	38.40 ± 1.43 ^a^	37.22 ± 0.86 ^a^
Aging+TP	76.12 ± 0.88 ^a,b^	60.80 ± 1.60 ^b^	55.68 ± 1.60 ^b^
Aging+saline	67.60 ± 1.07 ^a^	39.28 ± 1.20 ^a^	38.44 ± 1.06 ^a^
Aging+hGC	72.08 ± 0.93 ^a,b,c^	61.10 ± 1.09 ^b^	57.90 ± 1.27 ^b^
Diab+dmso	68.52 ± 0.71 ^a^	49.28 ± 1.24 ^a^	49.06 ± 0.80 ^a^
Diab+TP	75.39 ± 0.95 ^a,b^	64.36 ± 0.72 ^a,b^	65.84 ± 0.76 ^a,b^
Diab+saline	68.30 ± 0.84 ^a^	48.80 ± 0.86 ^a^	48.46 ± 0.98 ^a^
Diab+hGC	73.45 ± 0.95 ^a,b^	59.98 ± 0.59 ^b,c^	57.92 ± 1.04 ^b,c^

Notes: TP4/2 was administered for five days at a daily dose of 20 mg/kg (i.p.) and hCG was administered during five days at a daily dose of 20 IU/rat (s.c.). ^a^—the difference between the group Young+dmso and the other groups with the treatment DMSO or TP4/2+DMSO or between the group Young+saline and the other groups treated with saline or hCG+saline is significant at *p* < 0.05; ^b^—the difference between aging/diabetic rats without TP4/2 or hCG administration and aging/diabetic rats with TP4/2 or hCG administration is significant at *p* < 0.05; ^c^—the difference between the TP4/2- and hCG-treated rats within the young adult, aging and diabetic series is significant at *p* < 0.05. The data are presented as the M ± SEM, n = 5.

**Table 4 ijms-21-07493-t004:** The GlideScore, Emodel and Lipo/Coulomb/H-bond values resulting from molecular docking study of the compounds Org 43553 and TP4/2 into the transmembrane allosteric sites of rat LHCGR and TSHR.

Compound	GlideScore	Emodel	Lipo	Coulomb	H-Bond
Rat LHCGR					
Org 43553	−5.53	−51.36	−3.23	−4.87	−0.28
TP4/2	−5.07	−52.84	−2.47	−1.12	0.00
Rat TSHR					
Org 43553	−4.06	−57.83	−1.33	−6.44	0.00
TP4/2	−3.82	−38.71	−0.84	−6.66	−0.26

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
