# Peer review of "Comparative Study of the Steroidogenic Effects of Human Chorionic Gonadotropin and Thieno[2,3-D]pyrimidine-Based Allosteric Agonist of Luteinizing Hormone Receptor in Young Adult, Aging and Diabetic Male Rats"

_ijms, 2020, doi:10.3390/ijms21207493_

Round 1

Reviewer 1 Report

The manuscript by Bakhtyukov et al describes the newly designed low molecule weight agonist of LHCGR. Compound TP4/2 is a modification of previously described LMW agonist Org 43553. The effect of TP4/2 injection on testosterone production, the expression of steroidogenic genes expression, and its proliferative effects on germ cells in male rats was studied in three groups: adult males, diabetic, and aging males. The TP4/2 effects were compared with those of human CG hormone. According to authors, the Tp4/2 was less effective than hCG on testosterone production in first two days after injection but when injected for 5 days the effect was more pronounced with LMW agonist. The injection of TP4/2 led to an increase of seminiferous epithelium thickness and the number of spermatogonia and spermatocytes in intact adult, diabetic, and old males. The data are promising however there are several major limitations in this study:

  1. It is difficult to compare effects of hCG and TP4/2 compound as the route of delivery (SC vs IP), the vehicle (PBS vs DMSO), and active concentrations of two ligands were different.
  2. No PK or tissue bioavailability studies were performed with LMW agonist. No studies of ADME properties were performed. It is not clear whether TP4/2 is present in testis after IP injections
  3. The vehicle for TP4/2 compound is indicated as DMSO. DMSO is a highly cytotoxic chemical, it might induce inflammation and tissue damage in animals. The amount of injection is not indicated. To make the proper comparisons between hCG and TP4/2 all mice should have been injected with the same vehicles.
  4. The specificity of TP4/2 is not established. The authors argued that the absence of activation of thyroid hormones levels after injection and potential loss of predicted binding in modelling of TSHR and compound suggest that there is no cross-activation of this receptor. Again the presence of TP4/2 in target tissue, thyroid, was not established. The direct testing of receptor activation in cells expressing various GPCRs (such as HEK293 transfected with TSHR, FSH, ) will be more appropriate.
  5. There are few details of testicular sections analysis. What were criteria for the identification of various cell in the absence of specific markers staining? What about other germ cells, was there any change? Did you analyze all slides in blind fashion? How the tubules were selected for analysis? The measurements of the seminiferous epithelium are rather difficult to do as the thickness varies inside the tubule. The diameter of the tubules appearing as round ones on section would be more reliable.
  6. Why the 5-day treatment scheme was selected? Can authors provide references to similar studies if available?

Some other comments:

Line#

  1. “In diabetic and aging testes, TP4/2 restored spermatogenesis” it is impossible to conclude. The data showed an increased number of pre-meiotic cells. One would not see full restoration of spermatogenesis in 5 days.

Introduction: I do like the introduction. It is written well.

  1. Please show mean+/-SE for EC50

Fig.1. and other figures – Please make graphs the same size. In Fig.1 A is smaller than B

Fig.2 As mentioned above the differences in T level could be due to different PK/bioavailability/effective dosage of two ligands.

Fig. 2 Figures are not aligned.

Table 3 -5. What does F(2,147) mean? It appears that the explanations for Fig 3 (lines 282-285) were mixed up with something else. There is no Con-S3, Ag-S etc groups in this table.

Discussion: Too long.

346-348. Delete this paragraph. Looks like the instructions for authors.

352, 354. Change transmembrane channel to transmembrane domain

361-363. Delete this sentence, it is a repetition.

364-364. Can you clarify this statement: The low specificity of recombinant LH and hCG and urinary hCG in relation to the LH-regulated signaling cascades? What is urinary hCG?

  1. change capability to ability

377-379. there are no direct data (binding assays or any other functional assays) to confirm that TP4/2 interact with LHCGR in this paper. Presented data are indirect.

392-393. As noted above the data are indirect. Does TP4/2 able to activate TSHR downstream signaling pathways in cells expressing this GPCR? The authors did not perform any specificity analysis of TP4/2 with other GPCRs.

408-413. The authors are correct when the mention differences in PK/bioavailability. I would add the crucial differences in delivery (SC vs IP), dosage, and vehicle composition.

483-501. This can be shortened as no data presented in the paper substantiate these speculations.

Materials and Methods. Need more information.

  1. What was the volume of the injections?
  2. Should be Table S2

588-589. The description of this experiment is rather confusing. According to Table S2 the injections were done at 10-30am and the last blood draw was at 2pm. That is 3h 30min, not 5 hr as stated here.

  1. “hCG at the doses of ≥25 mg/kg and ≥50 IU/rat, respectively” This is confusing. What was the exact dose of TP4/2 and hCG used in the experiment?

699-700. Provide clear criteria for differentiation between spermatogonia vs pachytene spermatocytes. Confirm that all measurements were done in blind fashion.

Author Response

RESPONSE TO REVIEWER 1

The comparative study of steroidogenic effects of human chorionic gonadotropin and thieno[2,3-d]pyrimidine-based allosteric agonist of luteinizing hormone receptor in young adult, diabetic and aging male rats

New title: “Synthesis and characterization of thieno[2,3-d]pyrimidine-based allosteric agonist of LHCG receptor in young adult, aging and diabetic male rats”

Andrey A. Bakhtyukov, Kira V. Derkach, Maxim A. Gureev, Dmitry V. Dar’in, Viktor N. Sorokoumov, Irina V. Romanova, Irina Yu. Morina, Anna M. Stepochkina, and Alexander O. Shpakov

COMMON SECTION OF RESPONSE TO REVIEWERS

We are very grateful to the Reviewers for a detailed analysis of our article and for the comments made. We sincerely hope that the explanation, additions and changes made by us based on their comments and remarks have improved and expanded our article.

In accordance with the requirements of the reviewers, we significantly revised the article and made the following main changes and additions to it (in more detail, the changes and additions are presented below in the extended answers to questions and comments of the Reviewers 1 and 2).

  1. The title of the article has been changed. The new title: “Synthesis and characterization of thieno[2,3-d]pyrimidine-based allosteric agonist of LHCG receptor in young adult, aging and diabetic male rats”.
  2. The name of groups of animals has been changed, including the control animals that in revised varianr of the article are designated as Young (Young adult). The revised article included the animals groups treated with only DMSO (Young+dmso, Aging+dmso, and Diab+dmso), which were previously excluded by us, because they had characteristics similar to those in the animals groups injected with saline (in revised variant – Young+saline, Aging+saline, and Diab+saline). In accordance with this, the design of Figures and Tables was changed and the statistical processing of the results was carried out. The order of consideration of the series of animals has also been changed, which is now the following: Young, Aging and Diabetic, which is more logical.
  3. The Subsection 4.3 has been added to Methods on the selection of optimal doses and methods of drug (TP4/2 and hCG) delivery. The data on dose-response relationships for intraperitoneal hCG and subcutaneous TP4/2 were included into the Table S4. The justification has been given for the choice of the subcutaneous administration of gonadotropin and the intraperitoneal administration of TP4/2 for main experiments devoted to study of their steroidogenic effects in the in vivo conditions (Appendix 1 in the attached file).
  4. To prove the absence of significant differences in the steroidogenic effect of long-term TP4/2 and hCG administration in the conditions of different routes of their delivery, the additional experiments were carried out. The stimulating effects of 5-day intraperitoneal administration of hCG and 5-day subcutaneous administration of TP4/2 on testosterone levels in young adult male rats were studied. The data detailed in the Response to Reviewers (see below) show similar dynamics of the s.c. and i.p. administered hCG, as well as the s.c. and i.p. administration of TP4/2. However, i.p. hCG and, especially, s.c. TP4/2 have lower efficacy and greater variability in testosterone levels. This indicates the legitimacy of the choice of the used routes of drug administration to assess and compare their steroidogenic activity, namely the s.c. administration of gonadotropin and the i.p. administration of TP4/2.
  5. As part of proving the receptor specificity of TP4/2, we presented data on the absence of this effect on the basal and forskolin-, guanine nucleotide- and TSH-stimulated adenylyl cyclase activity in the rat thyroid membranes (an additional Table is presented in Supplement, as Table S2). The additional data on the lack of TP4/2 effect on the thyroid system (the in vitro experiments on the rat thyroid membranes, and the in vivo experiments on the investigation of TP4/2 effect on basal and TRH-stimulated thyroid hormones production in male rats) are presented and highlighted in a separate subsection in Results: “2.4. The effect of TP4/2 on the AC activity in thyroidal membranes and the thyroid hormones levels”. The details of the methodology for the in vivo experiments to assess the effect of TP4/2 pretreatment on the levels of thyroid hormones in rats are presented in subsection 4.4.
  6. Our study of the distribution of TP4/2 in the rat tissues using HPLC showed that this compound, when administered intraperitoneally to male rats, reaches both the testes (its main target) and the thyroid. As a result, the steroidogenic effect of TP4/2 is due to its ability to directly affect the Leydig cells of the testes. At the same time, the lack of TP4/2 effect on thyroidogenesis is not associated with its inability to enter the thyroid tissue, but is due to the inability of TP4/2 to interact with the TSH receptor. According to the data of chromatographic analysis, the content of TP4/2 in the testes and the thyroid gland of rats 1 and 3 hours after the intraperitoneal injection of the drug (20 mg/kg) was comparable (Appendix 2 in the attached file).
  7. Necessary additions and changes have been made in various sections, including Methods, Results and Discussion. In particular, the conditions of the experiments are indicated in more detail, changes have been made to the captions for Figures and Tables, the design of some Figures and Tables has been changed, and the changes have been made to the Discussion, including some paragraphs being shortened. The localization of germ cells in the seminiferous tubules is presented in the Appendix 3 (in the attached file).

According to requirement of Editor and Reviewers, all changes and additions to the text, including changes and additions in the Tables and Figures are highlighted in yellow.

In conclusion, we thank again for the Reviewers’ comments that allowed us to expand the article and introduce important additional material into it.

With best regards,

The Authors

Reviewer: 1

MAJOR REMARKS

1. It is difficult to compare effects of hCG and TP4/2 compound as the route of delivery (SC vs IP), the vehicle (PBS vs DMSO), and active concentrations of two ligands were different.

We are grateful to the reviewer for comments and a question about the route of delivery, the solvent and active concentrations of TP4/2 and hCG. It should be noted that both drugs differ greatly in their mechanisms of interaction with the LHCGR, their hydrophobicity, and, according to the data for the compound Org 43553, which has structural similarity with TP4/2, should have the differences in pharmacokinetics and pharmacodynamics.

At the stage of choosing and optimizing the route of delivery, we investigated different methods of TP4/2 administration, including i.p. and s.c. We have shown that s.c. administration is less effective than i.p. and, most importantly, results in significant variability in the testosterone-stimulating effect in rats. It can be assumed that this is due to lower bioavailability of TP4/2 when this compound is administered s.c. as compared to i.p. delivery.

Our dose-response studies showed that the dose of TP4/2, which caused steroidogenic effect, amounting to approximately 80% of the maximum (20 mg/kg), was similar when this compound administered intraperitoneally and subcutaneously, while the testosterone-stimulating effect in the case of s.c. administration was significantly lower. These preliminary obtained data on the s.c. administration of TP4/2 are added to the Table S4. As a result, we have chosen the i.p. route for TP4/2 administration.

In the case of hCG, we also studied both routes of delivery, and showed that i.p. administration is less effective and leads to more variable results as compared to s.c. administration. A dose-response study with s.c. and i.p. administration of hCG did not reveal a significant difference in the dose of gonadotropins, which causes a steroidogenic effect, which is 70-80% of the maximum (20 IU/rat). The preliminary obtained data on the i.p. administration of hCG are added to the Table S4. It is important to note that in a clinic, subcutaneous and also intramuscular administration, but not intraperitoneal, is widely used (Saal et al., 1991, PMID: 1712735; Weissman et al., 1996, PMID: 8908528; Trinchard-Lugan et al., 2002, PMID: 12470572; Chan et al., 2003, PMID: 14585876; Jin et al., 2015, PMID: 26170642). There is evidence that i.p. rather than s.c. administration leads to a number of side effects when administered to female mice (Lehtonen, Kankondi, 1987; PMID: 3656288). As a consequence, we have chosen the s.c. route of administration of hCG.

Thus, we have the different routes of administration for thienopyrimidine (i.p.) and gonadotropin (s.c.), based on the difference in their physicochemical and pharmacological properties.

To analyze the changes of the blood testosterone level during a five-day administration of drugs with both routes of delivery, from 17 to 21 September, we carried out the additiona experiments and studied the effect of s.c. administered TP4/2 (20 mg/kg) and i.p. administered hCG (20 IU/rat) on testosterone levels in four-month-old male rats. The results are presented in the Appendix 1 (see below), and show that there are no differences in the dynamics of testosterone levels with different methods of drug administration, but the effectiveness in the case of s.c. administration of TP4/2 is significantly lower, and the response of animals varies greatly. To a certain extent, this is also true for i.p. administration of hCG. Thus, the obtained data strengthen our arguments in the comparative analysis of i.p. administered TP4/2 and s.c. administered hCG.

The choice of the aprotic bipolar solvent DMSO for TP4/2 is due to the peculiarities of the physicochemical properties of this thieno[2,3-d]pyrimidine derivative and its analogous (Org 43553, etc.). At the same time, DMSO was unacceptable for gonadotropin. For details, see also the answer to the Comment 3.

It should be noted that in the presented article, our attention was focused on studying the mechanisms and targets of the steroidogenic effect of TP4/2, and the development of a dosage form went beyond the boundaries of our study.

SUMMARY:

1. A special section on the selection of doses and methods of administration has been added to the Methodology (the subsection 4.3).

4.3. The choice of the doses and delivery routs for TP4/2 and hCG

The choice of the doses for TP4/2 and hCG and the method of drug delivery were carried out on the basis of preliminary studies with Young adult (four-month-old) male Wistar rats.

To determine the optimal dose for different routes of drug administration, the TP4/2 and hCG were administered intraperitoneally and subcutaneously at the doses from 5 to 50 mg/kg for TP4/2 and from 5 to 100 IU/rat for hCG. At each dose, the volume of the TP4/2 solution was 200 μl (in DMSO), and the volume of the hCG solution was 200 μl (in saline). Control animals received either DMSO (in the experiments with TP4/2) or saline (in the experiments with hCG) in the same volume and with the same route of administration. The drugs or their solvents were administered at 10.00. In these experiments, the steroidogenic effect of hCG and TP4/2 was assessed by the blood T level during 5 hours after drugs administration (10.00–15.00) (Table S4).

In the case of TP4/2, the maximal stimulating effect on the T production was observed at the doses 25 mg/kg and above, while in the case of hCG, the effect was observed at the doses 50 IU/rat and above. In accordance with this, the doses were selected in which the steroidogenic effect of TP4/2 and hCG was 70-80% of its maximum value, and these doses were 20 mg/kg for TP4/2 and 20 IU/rat for hCG in both routes of administration.  

At the same time, in the case of s.c. administration, the maximum stimulating effect of TP4/2 was less pronounced and had a significant variability compared to i.p. administration, which, we believe, is due to the lower bioavailability of TP4/2 when administered subcutaneously. In the case of i.p. administration of hCG, the maximum steroidogenic effect of gonadotropin was lower than in the case of its s.c. administration, and had a significant variability. Based on these data, to study the steroidogenic effects of these drugs, we used i.p. administration of TP4/2 at a dose of 20 mg/kg and s.c. administration of hCG at a dose of 20 IU/rat.

2. Additional experiments were carried out on a five-day administration of hCG (i.p.) and TP4/2 (s.c.), and the data is presented in the Response to Reviewers (Appendix 1). The description of these experiments is presented in the subsection 4.3 (see above in Common Section of Response to Reviewers).

3. Table S4 has been added with the data on the selection of doses for a single injection of hCG (i.p.) and TP4/2 (s.c.). 

2. No PK or tissue bioavailability studies were performed with LMW agonist. No studies of ADME properties were performed. It is not clear whether TP4/2 is present in testis after IP injections

We thank the reviewer for question regarding the tissue distribution of TP4/2 in rats. In early works on thienopyrimidines, for the compound Org 43553 and its analogous, which have features of similarity to TP4/2 in the physico-chemical characteristics, some parameters of distribution in the body were established (van Koppen et al., 2008). In our study, we focus on the study of the steroidogenic effect of TP4/2, without special consideration of the bioavailability of this drug.

As part of the study of bioavailability and pharmacodynamics of TP4/2, we carried out the pilot research on the distribution of this compound in the blood and the testes and thyroid gland of rats using HPLC analysis.

The rats were injected with TP4/2 intraperitoneally, once, at a dose of 20 mg/kg, and after one or three hours the animals were decapitated under anesthesia (inhalation with 4–5 % (v/v) isoflurane) and the blood and the testes and thyroid gland were taken for further analysis. Control animals were treated with DMSO instead of drug.

The tissue samples (after weighing) were homogenized in the 50 mM Tris-buffer, pH 7.4, in the ratio 1:4. Then the homogenate was mixed with an ice-cooled acetonitrile in the ratio 1:2, and after that the obtained homogenate was centrifuged (12000 g, 15 min). The supernatant was used for HPLC analysis.

The plasma samples were mixed with ice-cold acetonitrile with the ratio 1:2 (v/v) and then centrifuged for 12000 g for 15 min and used for HPLC analysis.

We showed that the content of TP4/2 in the testes and thyroid 1 and 3 hours after drug i.p. administration of TP4/2 (20 mg/kg) was comparable, but at the same time significantly higher than in the blood (see Table in Appendix 2). After 3 hours, the content of TP4/2 in the tissues (but not in the blood) was decreased in comparison with the content of this drug 1 hour after administration. At the same time, the decrease of TP4/2 content in the thyroid gland was more pronounced than in the testes, which may be due to the specific binding of TP4/2 to LHCGRs and its retention in testicular cells (see Table in Appendix 2). TP4/2 was not detected in the blood and tissues of control male rats treated with DMSO alone.

Thus, we have established that within 1 hour after i.p. administration, TP4.2 is found not only in the blood, but also in the testes and the thyroid gland. The content of TP4/2 in both studied tissues one and three hours after its administration is comparable and indicates the intake and accumulation of TP4/2 in them. The data obtained are a good foundation for a more detailed and broad analysis of the distribution of TP4 in tissues both at various doses and modes of drug administration, and when studying drug clearance over time.

Specificity of the interaction of TP4/2 with LHCGR in the testes, and the lack of interaction of TP4/2 with TSHRs in the thyroid gland are also proven by our data obtained in the in vitro and in vivo experiments. Firstly, TP4/2 stimulates the basal AC activity in a dose-dependent manner in LHCGR-containing testicular membranes (the data is presented in the Fig. 1 and Table S1), and does not affect either the basal or TSH-stimulated AC activity in TSHR-containing thyroidal membranes (the data is added as the Table S2). Secondly, TP4/2 stimulates testicular steroidogenesis by different delivery routes, including i.p. and s.c. administration (the data in our article, including the additional data in the Table S4, as well as the results obtained with five-day s.c. administration of TP4/2, in the Response to Reviewers, Appendix 1). At the same time, an increase in the blood level of LH (Table 1) was not detected, which indicates the absence of a stimulating effect of TP4/2 on the hypothalamic and pituitary components of the HPG system. At the same time, there was no effect on the thyroid system, both basal and TRH-stimulated, which is important for assessing the absence of the effect of drug on the HPT system (Table S3).

SUMMARY

The abovementioned data indicate that the reason for the absence of the TP4/2 effect on the HPT system is not their inability to enter the thyroid gland tissues, but in its inability to influence the activity of TSH receptors.

3. The vehicle for TP4/2 compound is indicated as DMSO. DMSO is a highly cytotoxic chemical, it might induce inflammation and tissue damage in animals. The amount of injection is not indicated. To make the proper comparisons between hCG and TP4/2 all mice should have been injected with the same vehicles

Thieno[2,3-d]pyrimidine derivatives are slightly soluble in classical polar solvents - water and alcohol, but at the same time they are readily soluble in the bipolar aprotic solvent DMSO, which we used and was also used by other authors to dissolve chemically related compounds, including Org 43553. DMSO is used as a carrier for hydrophobic drugs and is one of the most common solvents used in biological research to dissolve hydrophobic substances in the in vivo and in vitro (Tsung et al., 2017, PMID: 28028757; Deol et al., 2019; PMID: 31191316), including the experiments with DMSO intraperitoneal administration to rodents (Shakoori et al., 2007, PMID: 17640304; Buigues et al., 2019, PMID: 30791852; Wang et al., 2020; PMID: 32445760).

The toxicity of DMSO is low. When this solvent is administered intraperitoneally, the LD50 in rats is 9.9 ml/kg, and when DMSO is administered intravenously, the LD50 in rats is 7.4 ml/kg (Kelava et al., 2011). According to Rubin, when administered intravenously to rats, the LD50 for DMSO is 5.2–8.1 g/kg (Rubin, 1975; PMID: 1093466). The volume of the DMSO solution we injected with or without TP4/2 dissolved in it (DMSO control) was 200 μl, which in terms of kg of animal body weight is approximately 0.6 ml/kg, which, according to Kelava and coauthors (Kelava et al., 2011), is 15 times below the LD50 value when DMSO is administered intraperitoneally.

We found no visible signs of deterioration in health in rats that received TP4/2 in DMSO or DMSO alone (i.p.) for five days. This is illustrated by the indicators of the control group receiving only one DMSO. There were no negative effects revealed in pilot experiments, when DMSO or DMSO-solution of thienopyrimidine derivatives was injected intraperitoneally for 10 days, which indicates the absence of a cumulative effect of DMSO with chronic administration.
This is consistent with the data of other authors who indicate that the toxicity of DMSO to animals is low, and DMSO is considered relatively safe and adverse reactions to DMSO are relatively mild and can occur only at high concentrations and certain routes of administration (Jacob et al., 2009; PMID: 19443933). It is important that DMSO has low systemic toxicity, but local toxicity can be manifested when using high concentrations.

Although there is controversy over the use of DMSO in clinical trials, DMSO is widely used in preclinical studies as a solvent for various drugs (Kahler, 2000, PMID: 11358355).
It is important to note that DMSO retains the biological activity and pharmacological properties of many drugs for a long period, increases their bioavailability and contributes to the achievement of a high intracellular concentration of drugs (Gnerre et al., 2000, PMID: 11123983; Kahler, 2000, PMID: 11358355).

Currently, we are carrying out a set of studies to optimize the dissolution of TP4/2 and associate good prospects with Cyclodextrins and their derivatives.

4. The specificity of TP4/2 is not established. The authors argued that the absence of activation of thyroid hormones levels after injection and potential loss of predicted binding in modelling of TSHR and compound suggest that there is no cross-activation of this receptor. Again the presence of TP4/2 in target tissue, thyroid, was not established. The direct testing of receptor activation in cells expressing various GPCRs (such as HEK293 transfected with TSHR, FSH, ) will be more appropriate

Our data on the specificity of TP4/2 are based on the following facts.

1. TP4/2 does not affect the basal and stimulated levels of thyroid hormones when administered to male rats at a dose that activates testicular steroidogenesis, and this excludes its ability to influence the functional activity of the thyroid gland in the in vitro conditions (see Table S3).

2. TP4/2 does not affect the basal and TSH-stimulated adenylyl cyclase activity in thyroid membranes isolated from the rat thyroid gland (the corresponding data was added to Supplemental Materials, as the Table S2, The basal and TSH-stimulated adenylyl cyclase activity in the rat thyroidal membranes in the presence of TP4/2). These membranes lack LHCGRs, which are the main targets for TP4/2, as well as for hCG and LH. This indicates, firstly, that TP4/2 does not activate TSH receptors and the adenylyl cyclase activity in the rat thyroid membranes, and, secondly, that it does not interfere with the functional interaction of hormone (TSH) with its receptor. Thus, it does not exhibit properties of either an agonist or neutral antagonist/inverse agonist of the TSH receptor.

The corresponding subsection has been added to Results section.

2.4. The effect of TP4/2 on the AC activity in thyroidal membranes and the thyroid hormones levels 

Since the structure of the allosteric site of LHCGR is similar to that of thyroid stimulating hormone receptor (TSHR), we studied the effect of TP4/2 on the AC activity in the rat thyroidal membranes in the in vitro conditions and on the baseline and thyrotropin-releasing hormone (TRH)-stimulated blood levels of thyroid hormones in young adult rats in the in vivo conditions. In the in vivo experiments, TRH was administered intranasally (300 μg/kg), as previously described [42].

No significant effect of TP4/2 (10-5 M) on the basal and forskolin-, GppNHp- and TSH-stimulated AC activity in the rat thyroidal membranes was found (Table S2). With a single administration to rats, TP4/2 at a dose of 20 mg/kg had no effect on the baseline and TRH-stimulated blood levels of free and total thyroxine and total triiodothyronine within 3.5 hours after TP4/2 administration (Table S3).

We also made additions to the Methods section, where we described the isolation of thyroid membranes (Heyma, Harrison, 1984, PMID: 6088581) and the determination of AC activity in them, both basal and stimulated by various agents.

3. Molecular docking data confirm our experimental results.

In this case, not only the energy characteristics of the interaction of TP4/2 with the inner surface of the transmembrane allosteric site, but also the lower hydrophobicity of the functional groups of the TP4/2 in comparison with the allosteric ligands for the TSH receptor are important. This does not allow the TP4/2 to penetrate into this site through the external entrance, which in the TSH receptor is formed by more hydrophobic amino acid residues in comparison with the LHCGR.

5. There are few details of testicular sections analysis. What were criteria for the identification of various cell in the absence of specific markers staining? What about other germ cells, was there any change? Did you analyze all slides in blind fashion? How the tubules were selected for analysis? The measurements of the seminiferous epithelium are rather difficult to do as the thickness varies inside the tubule. The diameter of the tubules appearing as round ones on section would be more reliable

Testicular staining with hematoxylin and eosin (H&E) is a recognized and widely used method for identification the different components located in the wall of the convoluted tubules and in the connective tissue between them. No specific markers are needed to identify them.

According to histological atlases (for example, interactive histological atlas:  http://www.ujaen.es/investiga/atlas/atlas_ingles/index.htm, see yellow square) or methodological guide (http://www.oecd.org/chemicalsafety/testing/40638492.pdf; ENDOCRINE DISRUPTION: A GUIDANCE DOCUMENT FOR HISTOLOGIC EVALUATION OF ENDOCRINE AND REPRODUCTIVE TESTS (Draft 3; Part 2).  2008. pp. 27-31; https://www.item.fraunhofer.de/reni/trimming/trimm.php?lan=en), the spermatogonia are located directly above the basement membrane. Spermatocytes are larger cells that are located above the spermatogonia. Pachytene spermatocytes have compacted chromatin or chromosomes in the nucleus or in the center of the cell, which indicates that the cells are at the stage of the primary and secondary meiotic division Appendix 3, see below). We did not count the number of other cells (the secondary spermatocytes, spermatids, and spermatozoa).

When carrying out morphometric studies, the generally accepted rules were observed, and two independent investigators measured the parameters in the blindfold conditions. Each researcher who carried out the evaluation did not initially know the identity of any of the specimens with respect to treatment and group assignment, which is evidence of blind analysis.

The corresponding additions have been made to subsection 4.15.

The tubules were selected for analysis in a cross section, i.e. tubules are rounded. The measurement of the height of the epithelium was carried out perpendicular to the basement membrane of about a smaller diameter, which corresponds to the counting method described earlier in work (Simas et al., 2017; PMID: 29285813).

6. Why the 5-day treatment scheme was selected? Can authors provide references to similar studies if available?

Previously, we studied different durations of administration of hCG and thienopyrimidine derivatives (3-7 days), and according to steroidogenic parameters we observed a similar dynamics of their change - with significant differences between hCG and low-molecular-weight agonists at the initial stage and with the convergence of their steroidogenic efficacy after three days of administration.

The five-day duration of treatment makes it possible to exclude acute short-term effects of drug administration and possible side effects from their long-term chronic administration, and, taking into account the use of rats, corresponds to the courses of gonadotropins in the correction of reproductive disorders and in assisted reproductive technologies. At the same time, we do not expect the launch of compensatory processes due to functional changes in the endocrine system within five days. Activating these processes would make the interpretation of the results more complex and ambiguous.

It should be noted, however, that we plan to carry out longer courses of therapy with thienopyrimidine derivatives in order to study both their effects during prolonged treatment and the long-term consequences after the end of such treatment.

Some other comments:

31 “In diabetic and aging testes, TP4/2 restored spermatogenesis” it is impossible to conclude. The data showed an increased number of pre-meiotic cells. One would not see full restoration of spermatogenesis in 5 days.

Thank you very much.

Phrase changed: “restored” replaced with “improved”.

TP4/2 improved spermatogenesis.

144 Please show mean+/-SE for EC50

Thank you very much.

The corresponding calculations were carried out and the values mean±SE for EC50 are presented in the text.

Fig.1. and other figures – Please make graphs the same size. In Fig.1 A is smaller than B

According to comments, we have made the appropriate changes to optimize the size of the figures (see Common Section of Response to Reviewers).  

Fig.2 As mentioned above the differences in T level could be due to different PK/bioavailability/effective dosage of two ligands.

We used the doses of hCG and TP4/2, which, with a single injection, had approximately 70-80% of the maximum steroidogenic effect. In relation to the different dynamics of the steroidogenic effects of gonadotropin and a LMW agonist, the differences in the bioavailability and pharmacokinetics of hCG and TP4/2 can have a significant effect on the magnitude and dynamics of the steroidogenic effect.

An addition is made in the Discussion (see also the Response below regarding lines 408-413):

This may be due to differences in bioavailability and pharmacokinetics of the TP derivatives and gonadotropins, their delivery route and vehicle composition [19],…”

Fig. 2 Figures are not aligned.

According to comments from Reviewer, we have made the appropriate changes to optimize the size of the figures.

Table 3 -5. What does F(2,147) mean? It appears that the explanations for Fig 3 (lines 282-285) were mixed up with something else. There is no Con-S3, Ag-S etc groups in this table.

We have made the required corrections. Part of the Notes in the Table 3 (old variant) was mistakenly copied from the Notes in the Table 5 (old variant), and this annoying error was corrected.

Discussion: Too long.

The text of the Discussion was shortened. The paragraph on the chaperone-like properties of thienopyrimidines has been significantly reduced.

346-348. Delete this paragraph. Looks like the instructions for authors.

We apologize for accidentally transferring the instructions for authors into the text when forming an article in Proforma. This paragraph has been deleted.

352, 354. Change transmembrane channel to transmembrane domain

The change has been made.

The term "channel" has been changed to "domain".

At the same time, we would like to note that the seven helical hydrophobic regions of the GPCR form a transmembrane structure of the channel-like type, which has much in common with classical ion channels. In particular, protons are transferred inside this "channel" in response to ligand binding and/or G-protein activation (exchange of GDP for GTP in guanine nucleotide-binding site of Galpha-subunit). At the same time, we agree that the term "channel" for the GPCR is ambiguous and should be deleted.

361-363. Delete this sentence, it is a repetition.

We agree that this is a repetition. The phrase has been deleted.

364-364. Can you clarify this statement: The low specificity of recombinant LH and hCG and urinary hCG in relation to the LH-regulated signaling cascades? What is urinary hCG?

Perhaps the phrase is not well constructed. It discusses both recombinant forms of gonadotropins with LH activity, which differ in their glycosylation from endogenous forms of LH and pituitary hCG circulating in the blood, and placental hCG, which is isolated from the urine of pregnant women (urinary hCG), which also differs from endogenous LH in activity and specificity for extracellular targets.

The phrase has been changed and only "LH and hCG preparations" are left.

A similar change is also made in the Introduction: “…use of gonadotropin preparations leads…”

378 change capability to ability

Made the required correction, “capability” replaced by “ability”

377-379. there are no direct data (binding assays or any other functional assays) to confirm that TP4/2 interact with LHCGR in this paper. Presented data are indirect.

We agree with the reviewer that we have not shown direct binding to the LHCGR, and our data on the interaction of TP4/2 with this receptor are indirect. As a result, the phrase has been changed. Emphasis is placed on TP4/2-induced stimulation of LHCGR-dependent processes.

Our experimental results have confirmed the ability of TP4/2 to stimulate LHCGR-dependent testicular steroidogenesis

392-393. As noted above the data are indirect. Does TP4/2 able to activate TSHR downstream signaling pathways in cells expressing this GPCR? The authors did not perform any specificity analysis of TP4/2 with other GPCRs.

Our data (no effect of TP4/2 on the basal and TSH-stimulated adenylyl cyclase in thyroid membranes in the in vitro conditions, Table S2, and the lack of TP4/2 effect on the basal and TRH-stimulated activity in the in vivo conditions) quite clearly indicate the absence of TP4/2 effect on TSH receptors and TSH-mediated physiological response. At the same time, we completely agree with the reviewer that, in the absence of direct evidence, it is more correct to discuss the effect of TP4/2 on TSH receptor-dependent processes and regulations than to focus the interaction of this low-molecular-weight compound with the TSH receptor. As a result, the corresponding phrase has been changed.

These data indicate the specificity of TP4/2 to the LHCGR- but not TSHR-dependent regulations, and largely exclude the negative effect of TP4/2 on the thyroid system”. 

408-413. The authors are correct when the mention differences in PK/bioavailability. I would add the crucial differences in delivery (SC vs IP), dosage, and vehicle composition.

We thank the Reviewer for the comment, and we include the differences in the vehicle composition and the route of delivery in the explanation of the differences between hCG and TP4/2.

This may be due to differences in bioavailability and pharmacokinetics of the TP derivatives and gonadotropins, their delivery route and vehicle composition [19],…”

483-501. This can be shortened as no data presented in the paper substantiate these speculations.

We fully agree with the Reviewer that information about the ability of a low-molecular-weight agonist to stabilize intracellular and defective forms of LHCGR is not supported by the data of the article. As a result, the paragraph, in accordance with the recommendations of the reviewer, is greatly shortened.

The fact that steroidogenic effect of TP4/2 slowly but gradually increase when it is administered for five days may be due to their ability to penetrate the plasma membrane of the Leydig cells and bind to “immature” forms of LHCGR localized intracellularly, such as was demonstrated for Org 43553 by other authors [37]. Thus, the TP derivatives can function as LMW chaperones for LHCGR. This may explain the fact that in T1DM, when the post-translational processing of testicular proteins is impaired and a significant part of them are modified by glucose residues, TP4/2 retains the ability to activate LHCGR and is not inferior to hCG, despite the lower affinity for the receptor.

Materials and Methods. Need more information.

Information was added in accordance with the comments of the reviewers.

579 What was the volume of the injections?

In accordance with the recommendations of the reviewer, an addition was made regarding the volumes of solutions containing hCG and TP4/2. Similar volumes were used in all experiments, as well as in the administration of saline and DMSO to the control groups.

, and the volume of the hCG solution was 200 μl (saline), and the volume of the TP4/2 solution was 200 μl (DMSO).”

580 Should be Table S2

The reference on the Table S3 (in the revised variant Table S4) is correct; it indicates the choice of the doses for TP4/2 and hCG, while the Table S2 (in the revised variant Table S3) corresponds to experiments on the effect of TP4/2 on the blood thyroid hormones levels.

588-589. The description of this experiment is rather confusing. According to Table S2 the injections were done at 10-30am and the last blood draw was at 2pm. That is 3h 30min, not 5 hr as stated here.

In the case of assessing the effect on thyroid hormones status, TP4/2 was injected at 10.30 (10.30 AM), then TRH or saline was injected at 11.00 (11 AM) and blood thyroid hormone levels were measured at 12.30 (12.30 AM) and 14.00 (2 PM), 2 and 3.5 hours after TP4/2 administration. This was a separate series of experiments, as a result of which an addition was made to the Methods (see subsection 4.4).

4.4. The estimation of thyroid hormone levels in TP4/2-treated rats 

To assess the effect of TP4/2 on the basal and TRH-stimulated blood levels of free (fТ4) and total (tТ4) thyroxine and total triiodothyronine (tТ3), the experiments were carried out on the young adult male rats. The blood levels of thyroid hormones were measured 2 and 3.5 hours after the pretreatment of the animals using i.p. injected TP4/2 in a single dose of 20 mg/kg. TRH (300 μg/kg, 10 μL) or saline (10 μL) was administered intranasally 30  min after pretreatment with TP4/2. The details of the experiment are given in the caption to the Table S3.

When evaluating the steroidogenic effect, the time of administration of the drugs every day was 10.00 (10 AM), and then within five hours (from 10.00 to 15.00; 10AM–3PM), the testosterone level was assessed. The same was the case when determining the optimal dose of drugs (Table S4).

591 “hCG at the doses of ≥25 mg/kg and ≥50 IU/rat, respectively” This is confusing. What was the exact dose of TP4/2 and hCG used in the experiment?

Appropriate corrections have been made.

+699-700. Provide clear criteria for differentiation between spermatogonia vs pachytene spermatocytes. Confirm that all measurements were done in blind fashion.

According to histological atlases and morphological investigations, the spermatogonia are located directly above the basement membrane. Spermatocytes are larger cells that are located above the spermatogonia. Pachytene spermatocytes have compacted chromatin or chromosomes in the nucleus or in the center of the cell (Appendix 3, see below).

When carrying out morphometric studies, the generally accepted rules were observed, and two independent investigators measured the parameters in the blindfold conditions. Each researcher who carried out the evaluation did not initially know the identity of any of the specimens with respect to treatment and group assignment, which is evidence of blind analysis.

The corresponding additions have been made to subsection 4.15.

Reviewer 2 Report

This MS by AA Bakhtyukov et al deals with the analysis of in vitro, in vivo and in silico properties of a new synthetic allosteric ligand of the  LHCG receptor, developped by the authors and named TP4/2.  Its properties are analyzed in this work, in comparison with hCG, in young, aged and diabetic rats (in vivo) or on cells from such animals (in vitro).

General comments

This paper is sound and very interesting scientifically and in view of its expected clinical developments in human reproduction. 

Nevertheless, the description of the experiments is confusing due to unappropriate terminology.

Indeed, the authors study the effects of TP4/2 ligand and hCG in young, aged and diabetic rats respectively. But they name the young rats « controls » :  « con-S1 »,  « con-S2 » and, « con-S3 » in the respective experiments with young, diabetic and aged rats respectively. The « con-S2 » and, « con-S3 » are not useful in the second and third experiments and introduce confusion with the true controls in these experiments that they call Diab-S and Ag-S . For the sake of simplification and ease of reading I would propose the disposition and names  shown in the table below.

The TP4/2 ligand and hCG are injected to rats in different solvants (DMSO and saline respectively) and through different routes ( ip and sc respectively). It would be important to check that it is not the reason for the differences observed between the kinetics of the responses. These data should be introduced as the genuine controls for each group as also shown in the table.

Finally I feel it would be better to present the two physiological conditions Young and Aged before the pathological one.

saline sc

Young

hCG sc

DMSO ip

TP ip

saline sc

Aged

hCG

DMSO ip

TP

saline sc

Diabetic

hCG

DMSO ip

TP

Specific  comments

  • In my opinion, the title is too long although it does not describe the whole content of the paper. I would propose : Synthesis and characterization of thienol…-based agonist of LHCG receptor in young adult, aging and diabetic male rats.
  • Splitting of tables should be avoided for the ease of reading
  • The symbols and text in the figures are too small and difficult to differenciate.
  • Fig 1 : Replace con-S1, Ag-S and Diab-S by Young, Aged and Diab. Increase size of the left panel.
  • Fig 2 : Indicate in the figure A :Young, B :Diab, C :Aged. Eliminate the names in the boxes and put the names of treatments (saline, TP, hCG) near the corresponding curves.
  • Fig 2 : What was the time of injection ? at 0h or 10h ? It is difficult to understand.
  • Fig 2 : Eliminate the curves con-S2 and con-S3 (see above) in both the left and right panels
  • Fig 3 and 4 : Same remarks as for Fig 2.
  • Lines 346-348 seem to come from a previous review by another reviewer.
  • Line 354 : add a reference concerning the GPCR transmembrane orthosteric sites.
  • Lines 380-381 : It should not be stated that the LHCGR and TSHR allosteric sites are close together. In fact the TSHR exhibits an amino-acid sequence that ressembles that of the recognized LHCGR allosteric site. This does not imply that it also plays a similar role. And in fact the authors clearly demonstrate that it is not the case.
  • In my opinion, it would also be interesting to consider in the future, the possible synergies between glycoprotein hormones and TP4/2. Would it work in hypox animals ?
  • Is there a possible direct effect of TP4/2 on membrane fluidity and raft formation ?

Author Response

RESPONSE TO REVIEWER 2

The comparative study of steroidogenic effects of human chorionic gonadotropin and thieno[2,3-d]pyrimidine-based allosteric agonist of luteinizing hormone receptor in young adult, diabetic and aging male rats

New title: “Synthesis and characterization of thieno[2,3-d]pyrimidine-based allosteric agonist of LHCG receptor in young adult, aging and diabetic male rats”

Andrey A. Bakhtyukov, Kira V. Derkach, Maxim A. Gureev, Dmitry V. Dar’in, Viktor N. Sorokoumov, Irina V. Romanova, Irina Yu. Morina, Anna M. Stepochkina, and Alexander O. Shpakov

COMMON SECTION OF RESPONSE TO REVIEWERS

We are very grateful to the Reviewers for a detailed analysis of our article and for the comments made. We sincerely hope that the explanation, additions and changes made by us based on their comments and remarks have improved and expanded our article.

In accordance with the requirements of the reviewers, we significantly revised the article and made the following main changes and additions to it (in more detail, the changes and additions are presented below in the extended answers to questions and comments of the Reviewers 1 and 2).

  1. The title of the article has been changed. The new title: “Synthesis and characterization of thieno[2,3-d]pyrimidine-based allosteric agonist of LHCG receptor in young adult, aging and diabetic male rats”.
  2. The name of groups of animals has been changed, including the control animals that in revised varianr of the article are designated as Young (Young adult). The revised article included the animals groups treated with only DMSO (Young+dmso, Aging+dmso, and Diab+dmso), which were previously excluded by us, because they had characteristics similar to those in the animals groups injected with saline (in revised variant – Young+saline, Aging+saline, and Diab+saline). In accordance with this, the design of Figures and Tables was changed and the statistical processing of the results was carried out. The order of consideration of the series of animals has also been changed, which is now the following: Young, Aging and Diabetic, which is more logical.
  3. The Subsection 4.3 has been added to Methods on the selection of optimal doses and methods of drug (TP4/2 and hCG) delivery. The data on dose-response relationships for intraperitoneal hCG and subcutaneous TP4/2 were included into the Table S4. The justification has been given for the choice of the subcutaneous administration of gonadotropin and the intraperitoneal administration of TP4/2 for main experiments devoted to study of their steroidogenic effects in the in vivo conditions (Appendix 1 in the attached file).
  4. To prove the absence of significant differences in the steroidogenic effect of long-term TP4/2 and hCG administration in the conditions of different routes of their delivery, the additional experiments were carried out. The stimulating effects of 5-day intraperitoneal administration of hCG and 5-day subcutaneous administration of TP4/2 on testosterone levels in young adult male rats were studied. The data detailed in the Response to Reviewers (see below) show similar dynamics of the s.c. and i.p. administered hCG, as well as the s.c. and i.p. administration of TP4/2. However, i.p. hCG and, especially, s.c. TP4/2 have lower efficacy and greater variability in testosterone levels. This indicates the legitimacy of the choice of the used routes of drug administration to assess and compare their steroidogenic activity, namely the s.c. administration of gonadotropin and the i.p. administration of TP4/2.
  5. As part of proving the receptor specificity of TP4/2, we presented data on the absence of this effect on the basal and forskolin-, guanine nucleotide- and TSH-stimulated adenylyl cyclase activity in the rat thyroid membranes (an additional Table is presented in Supplement, as Table S2). The additional data on the lack of TP4/2 effect on the thyroid system (the in vitro experiments on the rat thyroid membranes, and the in vivo experiments on the investigation of TP4/2 effect on basal and TRH-stimulated thyroid hormones production in male rats) are presented and highlighted in a separate subsection in Results: “2.4. The effect of TP4/2 on the AC activity in thyroidal membranes and the thyroid hormones levels”. The details of the methodology for the in vivo experiments to assess the effect of TP4/2 pretreatment on the levels of thyroid hormones in rats are presented in subsection 4.4.
  6. Our study of the distribution of TP4/2 in the rat tissues using HPLC showed that this compound, when administered intraperitoneally to male rats, reaches both the testes (its main target) and the thyroid. As a result, the steroidogenic effect of TP4/2 is due to its ability to directly affect the Leydig cells of the testes. At the same time, the lack of TP4/2 effect on thyroidogenesis is not associated with its inability to enter the thyroid tissue, but is due to the inability of TP4/2 to interact with the TSH receptor. According to the data of chromatographic analysis, the content of TP4/2 in the testes and the thyroid gland of rats 1 and 3 hours after the intraperitoneal injection of the drug (20 mg/kg) was comparable (Appendix 2 in the attached file).
  7. Necessary additions and changes have been made in various sections, including Methods, Results and Discussion. In particular, the conditions of the experiments are indicated in more detail, changes have been made to the captions for Figures and Tables, the design of some Figures and Tables has been changed, and the changes have been made to the Discussion, including some paragraphs being shortened. The localization of germ cells in the seminiferous tubules is presented in the Appendix 3 in the attached file.

According to requirement of Editor and Reviewers, all changes and additions to the text, including changes and additions in the Tables and Figures are highlighted in yellow.

In conclusion, we thank again for the Reviewers’ comments that allowed us to expand the article and introduce important additional material into it.

With best regards,

The Authors

Reviewer: 2

MAJOR REMARKS

1. This paper is sound and very interesting scientifically and in view of its expected clinical developments in human reproduction. 

Nevertheless, the description of the experiments is confusing due to unappropriate terminology.

Indeed, the authors study the effects of TP4/2 ligand and hCG in young, aged and diabetic rats respectively. But they name the young rats « controls » :  « con-S1 »,  « con-S2 » and, « con-S3 » in the respective experiments with young, diabetic and aged rats respectively. The « con-S2 » and, « con-S3 » are not useful in the second and third experiments and introduce confusion with the true controls in these experiments that they call Diab-S and Ag-S . For the sake of simplification and ease of reading I would propose the disposition and names  shown in the table below.

Thanks a lot for the comment, which contributed to a significant improvement in the structure of the article.

We fully agree with the opinion of the Reviewer on the optimization of the used control groups, including the exclusion of the Con-S2 and Con-S3 groups from consideration. It should be noted that during the preparation of the article, we excluded from consideration the groups treated with DMSO, since they did not show any significant changes in steroidogenic function and the tested parameters did not differ from the groups Con-S1, Ag-S and Diab-S, respectively. However, taking into account the requirements of both reviewers, we present data on DMSO-treated groups, excluding data on RT-PCR, where we did not study all control groups, based on our early data on the absence of the effect of five-day i.p. administration of DMSO on the indicators of steroidogenesis. Regarding PCR, it should be noted that in each series, the Young+saline group was selected as a control for which the comparison procedure was performed. The gene expression in the Young+saline group was taken as 100% (1.0). The comparison procedure is described in Methods.

All the necessary changes have been made to the structure of the article, to the Figures and Tables, and are also presented in the Figure for the design of the experiment.

2. The TP4/2 ligand and hCG are injected to rats in different solvents (DMSO and saline respectively) and through different routes ( ip and sc respectively). It would be important to check that it is not the reason for the differences observed between the kinetics of the responses. These data should be introduced as the genuine controls for each group as also shown in the table.

Thanks a lot for the comment.

We have added the data on steroidogenesis and morphology of the seminiferous tubules into the article for the animals treated with only DMSO. The groups Animals+DMSO were used for TP4/2-treated animals, and the groups Animals+saline were used for hCG-treated rats. The Animals+DMSO groups were previously excluded from the data presentation, so as not to complicate the structure of the article.

The data on the selection of doses not only for intraperitoneal administration of TP4/2 and subcutaneous administration of hCG, but also for subcutaneous administration of TP4/2 and intraperitoneal administration of gonadotropin are presented. These data are included in Table S4 and discussed in the Methods (subsection 4.3). We show that the subcutaneous administration of TP4/2 is not very effective, and the intraperitoneal administration of hCG is less effective than the subcutaneous administration. At the same time, the concentrations of TP4/2 and hCG that give 70-80% of the maximum testosterone-stimulating effect are similar (for details, see the discussion in the Common section in Response to Reviewers and in the responses to the comments of Reviewer 1, see above).

Along with this, additional experiments were carried out to evaluate five-day subcutaneous administration of TP4/2 and intraperitoneal administration of hCG on blood testosterone levels. The data obtained are presented in the Appendix 1, and show the fact that the dynamics of the steroidogenic effects of hCG and TP4/2 with different methods of drug administration is similar, but the efficiency is significantly differed. See also subsection 4.3.

3. Finally I feel it would be better to present the two physiological conditions Young and Aged before the pathological one.

We completely agree with the comment that it is better to first consider physiological conditions (age-related changes in the reproductive system), and only then pathological ones. As a result, in all sections of the article, we changed the order of presentation of data on diabetic and aging groups of male rats.

The order used is young-aging-diabetic.

Some other comments:

In my opinion, the title is too long although it does not describe the whole content of the paper. I would propose : Synthesis and characterization of thienol…-based agonist of LHCG receptor in young adult, aging and diabetic male rats.

Thank you for your comment and suggestion for improving the title of the article. We agree and changed the title of the article in accordance with your recommendations. The edited title: “Synthesis and characterization of thieno[2,3-d]pyrimidine-based allosteric agonist of LHCG receptor in young adult, aging and diabetic male rats

Splitting of tables should be avoided for the ease of reading

This remark was taken into account.

The symbols and text in the figures are too small and difficult to differenciate

This remark was taken into account.

Fig 1 : Replace con-S1, Ag-S and Diab-S by Young, Aged and Diab. Increase size of the left panel

The design of the Figures and Tables in the revised version of the article has been changed. They take into account all the comments. Additionally, the group titles in the revised version of the article were changed, and the groups with the administration of DMSO along were added to the Figures and Tables. These groups were previously excluded from consideration and were not presented in the original version of the article.

Fig 2 : Indicate in the figure A :Young, B :Diab, C :Aged. Eliminate the names in the boxes and put the names of treatments (saline, TP, hCG) near the corresponding curves

The group titles in the revised version of the article were changed, and the DMSO-treated groups were added (see response above).

Fig 2 : What was the time of injection ? at 0h or 10h ? It is difficult to understand

When evaluating the steroidogenic effect, the time of administration of the drugs every day was 10.00 (i.e. 10 AM), but not 0.00. Then, within five hours (from 10.00) – at 11.00, 13.00 and 15.00 (11 AM, 1 PM and 3PM) (the first day of administration, Fig. 2) or three hours after administration – at 13.00 (1 PM) (2-5 days, Fig. 3), the testosterone level was assessed. The same was the case when determining the optimal dose of drugs (Table S4).

Fig 2 : Eliminate the curves con-S2 and con-S3 (see above) in both the left and right panels

The group titles in the revised version of the article were changed, and the group Con-S2 and Con-S3 were excluded from consideration.

Fig 3 and 4 : Same remarks as for Fig 2

Figure captions in the revised version of the article have been changed. They take into account all the comments.

Lines 346-348 seem to come from a previous review by another reviewer

We apologize for accidentally transferring the instructions for authors into the text when forming an article in Proforma. This paragraph has been deleted.

Line 354 : add a reference concerning the GPCR transmembrane orthosteric sites

The phrase about the transmembrane orthosteric site in the LHCGR and glycoprotein pituitary hormones has been changed, and, according to the recommendation of the reviewer, three references have been made regarding the study of the structure, interaction pattern and molecular modeling of the transmembrane domain of such receptors.

The added references:

Troppmann et al., 2013, PMID: 23686864; Chantreau et al., 2015, PMID: 26545118; Kleinau et al., 2017, PMID: 28484426

It can be assumed that in evolutionary precursors of the LHCGR and other receptors of pituitary glycoprotein hormones, the transmembrane orthosteric site, due to structural changes in the transmembrane domain, including the replacement of the proline residue in the fifth transmembrane region, which is highly conserved in Class A GPCRs [Troppmann et al., 2013; Chantreau et al., 2015; Kleinau et al., 2017], has lost the ability to bind hormonal regulators and, as a result, transformed into an allosteric site.”

Lines 380-381 : It should not be stated that the LHCGR and TSHR allosteric sites are close together. In fact the TSHR exhibits an amino-acid sequence that ressembles that of the recognized LHCGR allosteric site. This does not imply that it also plays a similar role. And in fact the authors clearly demonstrate that it is not the case.

We agree with the reviewer's comment regarding the functional differences between the LHCGR and TSHR allosteric sites. In accordance with this, the phrase has been rearranged. It indicates the presence of features of similarity in the structure of these sites, which indicates the possibility of structurally similar ligands to cross-link with them and requires testing of these ligands for specificity to LHCGR- and TSHR-dependent cascades, which we carried out in our study. The modified phrase is shown below.

Despite the functional differences, the structural organization of the transmembrane allosteric sites LHCGR and TSHR have features of similarity [50]”.

In my opinion, it would also be interesting to consider in the future, the possible synergies between glycoprotein hormones and TP4/2. Would it work in hypox animals?

This is an extremely important problem and a promising prospect in the use of thienopyrimidine derivatives, and we are now actively investigating this problem. Using our other derivative, we also showed the ability of pretreatment of male rats with it to potentiate the steroidogenic effect of later administered low-dose gonadotropin (7.5 IU/rat). We hope that for TP4/2 it will turn out even better. This is perhaps the most popular of the possible applications of thienopyrimidine derivatives in assisted reproductive technologies, since it allows reducing the dose of hCG with controlled induction of ovulation and preventing ovarian hyperstimulation syndrome and the potential negative epigenetic effects of high-dose gonadotropin. Your proposal to study hypoxic animals, including rats with testicular ischemia/reperfusion injury, is also very interesting, and we will definitely consider it, especially since the chaperone-like effect of thienopyrimidines may appear here, which we assume in TP4/2 in accordance with our data in diabetes. With certain limitations, aging and severe streptozotocin type 1 diabetes can also be considered as testicular hypoxia-like states.

Is there a possible direct effect of TP4/2 on membrane fluidity and raft formation?

Lipid rafts and their associated signaling complexes play an important role in sperm maturation and control sperm fertility (Watanabe et al., 2017, PMID: 28425215). Despite the hydrophobic nature and lipophilicity, it is difficult to assume (although cannot be excluded) a direct effect of thienopyrimidines, including TP4, on membrane fluidity and the formation of lipid rafts. On the other hand, their indirect influence is possible, which can be realized in two ways. First, thienopyrimidine, through the activation of LHR-dependent cascades, can influence the effector systems involved in the regulation of the properties of the lipid matrix, the distribution and accumulation of cholesterol and glycosphingolipids. The second may involve the effect of thienopyrimidines on oligomerization and association of LHCGR with membrane structures, which can change the structural organization of signalosomes in lipid rafts. All this requires experimental verification, including taking into account the chaperone-like properties of thienopyrimidines and their possible effect on LHR translocation and their complex formation.

Round 2

Reviewer 2 Report

The authors have thoroughly answered all the points I raised and I am very satisfied with this amended version.

Author Response

Dear Reviewer,

We are grateful for the detailed consideration of the revised version of our manuscript and its positive assessment.

We agree that the ending of the phrase about blind analysis may be redundant, and therefore removed the second part of the phrase from the text (lines 755-756).

The old variant “Each researcher who carried out the evaluation did not initially know the identity of any of the specimens with respect to treatment and group assignment, which is evidence of blind analysis” was replaced with the new variant “Each researcher who carried out the evaluation did not initially know the identity of any of the specimens with respect to treatment and group assignment”.

With best regards,

The Authors